# Structural insights into Pot1-ssDNA, Pot1-Tpz1 and Tpz1-Ccq1 Interactions within fission yeast shelterin complex

Hong Sun[1,2,3☯], Zhenfang Wu[4,5☯]*, Yuanze Zhou[6☯], Yanjia Lu[4,5], Huaisheng Lu[4,5], Hongwen Chen[1,3], Shaohua Shi[4,5], Zhixiong Zeng[7]*, Jian Wu[4,5]*, Ming Lei[4,5,8,9]*

**1** State Key Laboratory of Molecular Biology, CAS Center for Excellence in Molecular Cell Science, Shanghai Institute of Biochemistry and Cell Biology, Chinese Academy of Sciences, Shanghai, China, **2** School of Life Science and Technology, ShanghaiTech University, Shanghai, China, **3** University of Chinese Academy of Sciences, Chinese Academy of Sciences, Beijing, China, **4** Ninth People's Hospital, Shanghai Jiao Tong University School of Medicine, Shanghai, China, **5** Shanghai Institute of Precision Medicine, Shanghai, China, **6** National Key Laboratory of Crop Genetic Improvement, Huazhong Agricultural University, Wuhan, China, **7** Shandong Provincial Key Laboratory of Microbial Engineering, College of Bioengineering, Qilu University of Technology, Shandong, China, **8** State Key Laboratory of Oncogenes and Related Genes, Shanghai Jiao Tong University School of Medicine, Shanghai, China, **9** Key Laboratory of Cell Differentiation and Apoptosis of Chinese Ministry of Education, Shanghai Jiao Tong University School of Medicine, Shanghai, China

☯ These authors contributed equally to this work.

* zhenfwu@shsmu.edu.cn (ZW); zengzx@qlu.edu.cn (ZZ); wujian@shsmu.edu.cn (JW); leim@shsmu.edu.cn (ML)

**Data Availability Statement:** The atomic models of Pot1DBD-Tel18, Pot1OB3-Tpz1PIM and Ccq1TAD-Tpz1CBM were deposited in Protein Data

## Abstract

The conserved shelterin complex caps chromosome ends to protect telomeres and regulate telomere replication. In fission yeast *Schizosaccharomyces pombe*, shelterin consists of telomeric single- and double-stranded DNA-binding modules Pot1-Tpz1 and Taz1-Rap1 connected by Poz1, and a specific component Ccq1. While individual structures of the two DNA-binding OB folds of Pot1 (Pot1$_{OB1}$-GGTTAC and Pot1$_{OB2}$-GGTTACGGT) are available, structural insight into recognition of telomeric repeats with spacers by the complete DNA-binding domain (Pot1$_{DBD}$) remains an open question. Moreover, structural information about the Tpz1-Ccq1 interaction requires to be revealed for understanding how the specific component Ccq1 of *S. pombe* shelterin is recruited to telomeres to function as an interacting hub. Here, we report the crystal structures of Pot1$_{DBD}$-single-stranded-DNA, Pot1$_{372-555}$-Tpz1$_{185-212}$ and Tpz1$_{425-470}$-Ccq1$_{123-439}$ complexes and propose an integrated model depicting the assembly mechanism of the shelterin complex at telomeres. The structure of Pot1$_{DBD}$-DNA unveils how Pot1 recognizes *S. pombe* degenerate telomeric sequences. Our analyses of Tpz1-Ccq1 reveal structural basis for the essential role of the Tpz1-Ccq1 interaction in telomere recruitment of Ccq1 that is required for telomere maintenance and telomeric heterochromatin formation. Overall, our findings provide valuable structural information regarding interactions within fission yeast shelterin complex at 3' ss telomeric overhang.

Bank with accession codes 7CUH, 7CUI and 7CUJ, respectively.

**Funding:** This work was supported by grants from the Ministry of Science and Technology of China (2018YFA0107004 to M.L.), the National Natural Science Foundation of China (31930063 to M.L. and 31270787 to Z.Z.) and the Shanghai Municipal Education Commission—Gaofeng Clinical Medicine Grant Support (20181711 to J.W.). The funders had no role in study design, data collection and analysis, decision to publish, or preparation of the manuscript.

**Competing interests:** The authors have declared that no competing interests exist.

## Author summary

Telomeres, composed of repetitive DNA sequences and specialized proteins, are protective structures at the ends of linear chromosomes. The telomere structure is essential for the maintenance of genome integrity and stability, and telomere dysfunction has been linked to human development, aging, cancer and a variety of degenerative diseases. An evolutionarily conserved multiple-protein complex called shelterin plays versatile roles in telomere homeostasis regulation, end protection and heterochromatin establishment. However, the highly flexible nature of shelterin complex has greatly impeded our structural and functional understanding for this important complex. In fission yeast, structures of the shelterin dsDNA-binding protein subcomplex Taz1-Rap1 and the bridge subcomplex Tpz1-Poz1-Rap1 are available. Although individual OB-fold subdomains structures have been characterized, structural information about the complete $Pot1_{DBD}$ bound to telomeric repeats with spacers remains to be revealed. Here, by determining the crystal structures of the telomeric overhang binding $Pot1_{DBD}$-ssDNA, $Pot1_{372-555}$-$Tpz1_{185-212}$ and $Tpz1_{425-470}$-$Ccq1_{123-439}$ subcomplexes, we provide structural basis not only for the recognition of *S. pombe* degenerate telomeric sequences by Pot1, but also for the essential function of the Tpz1-Ccq1 interaction in Ccq1 recruitment to telomeres for telomere maintenance and telomeric heterochromatin formation. These findings provide an integrated model depicting the assembly mechanism of the shelterin complex at telomeres and its multiple roles in telomere biology.

## Introduction

Telomere, the specialized nucleoprotein capping structure at the end of eukaryotic chromosomes, is essential for genome integrity and stability [1–3]. Telomeric DNAs consist of short tandem G-rich repetitive sequences, and terminate in a 3' single-stranded (ss) G-rich overhang [4]. The 3' ssDNA overhang serves as the substrate for telomerase, a specialized reverse transcriptase that uses its intrinsic RNA component as the template to fully replicate the chromosome ends, providing a solution for the end-replication problem to ensure genome integrity [5–10].

In human cells, a six-protein complex called shelterin, composed of telomeric double-stranded (ds) DNA-binding proteins TRF1 and TRF2, ssDNA-binding protein POT1, and bridging factors RAP1, TIN2 and TPP1 [11], plays essential roles in telomere homeostasis regulation and telomere protection [12]. Similarly, fission yeast *Schizosaccharomyces pombe* also contains a six-protein complex that closely resembles the human shelterin, composed of Taz1, Rap1, Poz1, Tpz1, Pot1 and Ccq1 [13–15]. In *S. pombe* shelterin complex, Taz1 and Pot1 bind to telomeric dsDNA and ssDNA regions respectively, whereas Rap1, Poz1 and Tpz1 are the bridging factors [14,15]. As a specific component in *S. pombe* shelterin, Ccq1 associates with telomeres via an interaction with Tpz1 [16,17], functioning as a platform for protein complexes that are essential for telomere maintenance and telomeric heterochromatin formation [16–18]. *S. pombe* Tpz1 is the homolog of human TPP1 and plays versatile roles in telomere maintenance and regulation. First, Tpz1, together with Poz1 and Rap1, interacts with Taz1 and Pot1, forming a bridge between the ds and ss regions of telomeres [14,15]. Second, Tpz1 forms a subcomplex with Pot1 to bind and protect telomeric ssDNAs and to regulate telomere homeostasis [13,19]. Third, the Tpz1-Ccq1 subcomplex is required for telomerase recruitment and activation via an Rad3/Tel1-dependent interaction with telomerase subunit Est1 [16]. Finally, the Tpz1-Ccq1 subcomplex is also important for telomeric heterochromatin formation

by recruiting two heterochromatic complexes SHREC (Snf2-HDAC repressor complex) and CLRC (Clr4 methyltransferase complex) to telomeres [18,20]. Thus, Tpz1-centered interaction network plays key roles in telomere homeostasis, telomere end protection and heterochromatin formation at telomeres [16–18]. Structures of the *S. pombe* shelterin dsDNA-binding protein subcomplex Taz1-Rap1 and the bridge subcomplex Tpz1-Poz1-Rap1 have been reported [14,15,21]. However, structural information of the telomeric ssDNA-binding protein subcomplex Pot1-Tpz1-Ccq1 has yet to be revealed, hindering our understanding of the architecture and the function of the *S. pombe* shelterin complex.

S. pombe Pot1 shares a similar domain organization as human POT1, consisting of two oligonucleotide/oligosaccharide (OB) folds at the N-terminus that confer the ssDNA-binding activity (hereafter referred to as the DNA-binding domain of Pot1, $Pot1_{DBD}$) and a Tpz1-binding domain at the C-terminus (Fig 1A) [13,22,23]. The N-terminal two OB folds of human POT1 specifically recognize 10-nucleotide (nt) telomeric ssDNA (5'-TTAGGGTTAG-3') as a single functional unit [24]. In contrast, the two OB folds in *S. pombe* $Pot1_{DBD}$ ($Pot1_{OB1}$ and $Pot1_{OB2}$) are separated by a 23-amino-acid linker; $Pot1_{OB1}$ and $Pot1_{OB2}$ can individually bind telomeric ssDNA, recognizing 6-mer (Tel6, GGTTAC) and 9-mer (Tel9, GGTTACGGT) ssDNAs with distinct specificities, respectively. The crystal structures of the $Pot1_{OB1}$-Tel6 and $Pot1_{OB2}$-Tel9 complexes have been solved respectively [25,26]. In the $Pot1_{OB1}$-Tel6 structure, the ssDNA binds in a basic concave groove that is characteristic of OB-fold proteins. The $Pot1_{OB1}$-Tel6 structure reveals that DNA self-recognition contributes to the sequence specificity of $Pot1_{OB1}$ binding [26]. Structures of $Pot1_{OB2}$ with different ssDNA ligands reveal multiple binding modes of $Pot1_{OB2}$ that explain its nonspecific recognition of ssDNA [25]. Notably, the *S. pombe* telomeric sequence is irregular, in which the 5'-GGTTAC-3' core sequence are separated by 0–8 linker nucleotides [27,28]. Structural information about the complete $Pot1_{DBD}$ bound to telomeric repeats with spacer sequences still has yet to be revealed, hindering our understanding of how *S. pombe* Pot1 recognizes the irregular cognate telomeric sequence.

In this study, we determine the crystal structures of the $Pot1_{DBD}$-ssDNA, $Pot1_{372-555}$-$Tpz1_{185-212}$ and $Tpz1_{425-470}$-$Ccq1_{123-439}$ subcomplexes, providing structural basis for the recognition of *S. pombe* degenerate telomeric sequences by Pot1 and the essential function of the Tpz1-Ccq1 interaction in Ccq1-dependent telomere maintenance and telomeric heterochromatin formation. We propose an integrated model depicting how the *S. pombe* shelterin complex assembles and plays its roles at telomeres.

## Results

### Structure of the $Pot1_{DBD}$-ssDNA complex

To understand how *S. pombe* $Pot1_{DBD}$ can recognize and accommodate irregular ss telomeric repeats, we examined the interactions of $Pot1_{DBD}$ with nine telomeric ssDNAs that contains the canonical ssDNA sequences for both $Pot1_{OB1}$ and $Pot1_{OB2}$ with 0–8 linker nucleotides (Tel15-Tel23) (S1 Fig) [25,26]. Gel filtration chromatography analysis showed that $Pot1_{DBD}$ binds Tel15, Tel17 and Tel18 with a 1:1 stoichiometry (S1D, S1E and S1F Fig). Furthermore, isothermal calorimetry (ITC) analysis revealed that $Pot1_{DBD}$ recognizes all tested telomeric ssDNAs with high binding affinities (Figs 1B and S1G).

Next, we crystallized $Pot1_{DBD}$ in complex with Tel18 and determined the complex structure by molecular replacement at a resolution of 3.0 Å (S1 Table). The calculated electron density map allowed us to unambiguously trace most of the complex except for a disordered loop between $Pot1_{OB1}$ and $Pot1_{OB2}$ (residues 174–198) (Fig 1C). Electron density map for the entire ssDNA was observed (S2A Fig). The overall structure of the $Pot1_{DBD}$-Tel18 complex reveals that $Pot1_{OB1}$ and $Pot1_{OB2}$ pack in tandem to adopt an elongated conformation, and the

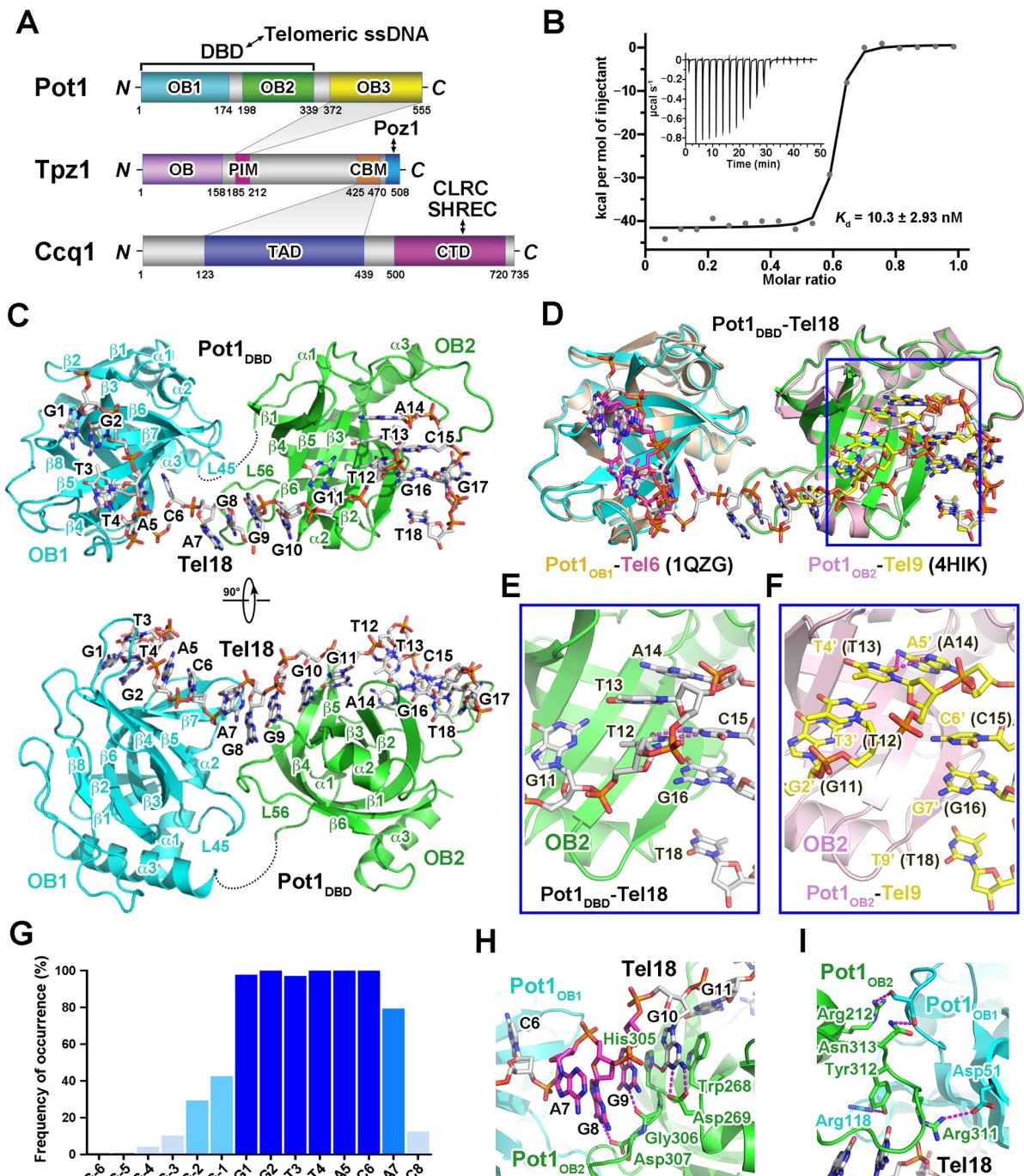

**Fig 1. Overview of the Pot1$_{DBD}$-Tel18 complex structure.** (A) Domain organization of *S. pombe* Pot1, Tpz1 and Ccq1. The shaded areas indicate the interactions between Pot1 and Tpz1 and between Tpz1 and Ccq1, respectively. PIM: Pot1-interacting motif of Tpz1; CBM: Ccq1-binding motif of Tpz1; TAD: Tpz1-associating domain of Ccq1; CTD: C-terminal domain of Ccq1. (B) ITC measurement of the interaction between Pot1$_{DBD}$ and Tel18. Inset: ITC titration data. (C) Overall structure of the Pot1$_{DBD}$-Tel18 complex in two orthogonal views. The OB1 and OB2 domains of Pot1$_{DBD}$ are colored in cyan and green, respectively. The disordered 25-residue loop (residues 174–198) between OB1 and OB2 is shown as a black dotted line. (D) Structural comparison of the Pot1$_{DBD}$-Tel18 complex with the Pot1$_{OB1}$-Tel6 and the Pot1$_{OB2}$-Tel9 sub-complexes. The Pot1$_{DBD}$-Tel18 complex structure is colored as in Fig 1C. Tel6 in Pot1$_{OB1}$-Tel6 and Tel9 in Pot1$_{OB2}$-Tel9 are colored in magenta and yellow, respectively. The major difference between Pot1$_{DBD}$-Tel18 and Pot1$_{OB2}$-Tel9 structures are highlighted in blue rectangle boxes. (E and F) Detailed analyses of the difference between Pot1$_{DBD}$-Tel18 (E) and Pot1$_{OB2}$-Tel9 (F) structures. Hydrogen bonds are shown as dashed magenta lines. (G) Experimentally determined occupancy frequency of each nucleotide position in *S. pombe* telomere sequence. (H) Detailed interactions between spacer nucleotides A7G8G9 in Tel18 and the surrounding residues of Pot1$_{DBD}$. Sidechains of residues important for the interactions are shown in stick models. (I) Details of the interactions between Pot1$_{OB1}$ and Pot1$_{OB2}$. Residues that mediate the interactions at the interface are shown in stick model form.

ssDNA-binding grooves of the two OB folds are connected with a kink at the OB1-OB2 interface (Figs 1C and S2B). The telomeric ssDNA Tel18 meanders along a long continuous groove on the surface of Pot1$_{DBD}$ in an 'S'-shaped conformation with its backbone exposed to solvent and its bases partially or completely buried in a solvent-excluded contact area of ~1,900 Å$^2$ (Figs 1C, S2B and S2C).

Close inspection of the Pot1$_{DBD}$-Tel18 interface reveals that G1-C6 and G10-T18 are respectively recognized by Pot1$_{OB1}$ and Pot1$_{OB2}$ almost in the same manner as in the Pot1$_{OB1}$-Tel6 and Pot1$_{OB2}$-Tel9 subcomplexes whose structures were previously determined (Fig 1C and 1D) [25,26]. The major difference is from nucleotides T12 and T13, which adopt distinct conformations in the two complexes (Fig 1D, 1E and 1F) [25]. In the Pot1$_{DBD}$-Tel18 complex, T12 and T13 stack together with the base of T12 forming two intramolecular hydrogen-bonding interactions with the base of C15 (Fig 1E). In contrast, in the Pot1$_{OB2}$-Tel9 subcomplex the base of T3' (T12 in the Pot1$_{DBD}$-Tel18 complex) undergoes a ~90° rotation away from C6' (C15) and mediates stacking interactions with both G2' (G11) and T4' (T13) (Fig 1F). The position of T4' (T13) is adjusted accordingly, leading to two hydrogen bonds between T4' (T13) and A5' (A14) (Fig 1F). Despite these local conformational changes, the ssDNA-binding register is maintained in both Pot1$_{OB2}$-Tel9 and Pot1$_{DBD}$-Tel18 complexes (Fig 1D) [25]. This is in accordance with previous studies that mutations of nucleotides T12 and T13 can be well accommodated by Pot1$_{OB2}$ with no detectable effect on ssDNA binding of Pot1$_{DBD}$ [25].

## Recognition of *S. pombe* degenerate telomeric ssDNA by Pot1

*S. pombe* telomeric DNA consists of multiple repeats of 5'-GGTTAC-3' core sequence with 0–8 linker nucleotides [27,28]. The crystal structure of the Pot1$_{DBD}$-Tel18 complex provides us a unique opportunity to understand how Pot1 could accommodate this degenerate telomeric sequence. In the Pot1$_{DBD}$-Tel18 complex, the Pot1$_{OB1}$ moiety recognizes G2T3T4 with high specificity, defining the binding register of the first telomere repeat (G1-C6) (Figs 1C, 1D, S2B and S2C) [26]. Compared with Pot1$_{OB1}$, Pot1$_{OB2}$ confers only moderate sequence specificity for nucleotides G11, G16 and T18 [29]. Nonetheless, this limited sequence specificity is able to define the binding register for the second telomere repeat (G10-C15) (Figs 1C, 1D, S2B and S2C) [29]. The well-aligned C15-G16-Trp223-G18-Tyr224 stack between Pot1$_{OB2}$ and the ssDNA further stabilizes the registered Pot1$_{DBD}$-Tel18 interface (S2D Fig).

Trinucleotide linker A7G8G9 in Tel18 is the most frequently occurring linker sequence in *S. pombe* telomeres (Fig 1G) [27,28]. This short nucleotide linker adopts a highly zigzagged conformation and is sandwiched between the two OB folds, with the bases of G8 and G9 stacking together on the plane formed by G10 and the side chain of His305 from Pot1$_{OB2}$ (Fig 1H). This continuous stacking conformation is stabilized by hydrogen-bonding interactions with highly conserved residues Asp269, Gly306 and Asp307 of Pot1$_{OB2}$ (Figs 1H and S3). The base of A7 flips away from the G8-G9 stack and makes no direct contact with Pot1$_{DBD}$ (Fig 1H). Based on the Pot1$_{DBD}$-Tel18 structure, we generated structural models of Pot1$_{DBD}$ bound to telomeric repeats with different spacers according to tight conformational and stereochemical constraints on both DNAs and proteins. Removal of G8 or the G8-G9 dinucleotide from the crystal structure, combined with some minor local conformational adjustments of the ssDNA, allowed us to generate the Pot1$_{DBD}$-ssDNA structural models with one- or two-nucleotide linker, which are almost identical to the Pot1$_{DBD}$-Tel18 complex structure (S4 Fig). Therefore, the conformation of Pot1$_{DBD}$ observed in the Pot1$_{DBD}$-Tel18 complex could explain how Pot1$_{DBD}$ binds two telomeric core repeats with one to three linker nucleotides (Figs 1B and S1G).

Pot1$_{DBD}$ can efficiently associate with Tel15, in which two telomeric core repeats are directly linked together with no linker nucleotide (S1D, S1G and S1H Fig) [19,30]. Structural

modeling reveals that $Pot1_{OB1}$ and $Pot1_{OB2}$ have to reorganize their positions to form a shorter ssDNA-binding groove to accommodate Tel15 (S4 Fig). Similarly, binding of ssDNA with a longer linker sequence (> 3 nt) would also induce the reorganization of the two OB folds of $Pot1_{DBD}$. Notably, these reorganizations inevitably should disengage the $Pot1_{OB1}$-$Pot1_{OB2}$ interface observed in the $Pot1_{DBD}$-Tel18 complex, which is only mediated by four electrostatic contacts and therefore makes limited contributions to the ssDNA binding (Fig 1I). Consistent with this idea, $Pot1_{OB1}$ and $Pot1_{OB2}$ are separated by a long 23-residue loop, that allows these two OB folds function as ssDNA-binding modules independently (Figs 1C and S4) [25,26]. This is in sharp contrast to the hydrophobic interface between OB1 and OB2 in human POT1, which can only function together as a single entity to recognize a regular human telomeric sequence TTAGGGTTAG [24]. A recent cryo-EM analysis has revealed alternative conformations of the two OB folds of human POT1 suggestive of its plasticity in DNA binding [31]. Here, we consider that the structurally separable $Pot1_{OB1}$ and $Pot1_{OB2}$ together with the long flexible loop between them endow *S. pombe* $Pot1_{DBD}$ with more flexibility capable of binding degenerate telomeric sequences. This is also supported by biochemical data that the binding affinity of Pot1 is not significantly affected by addition of spacer sequences (Figs 1B and S1G) [30,32].

## Crystal structure of the $Pot1_{OB3}$-$Tpz1_{PIM}$ complex

Similar to how human POT1 interacts with TPP1, the C-terminal portion of the *S. pombe* Pot1 ($Pot1_{372-555}$) mediates the interaction with Tpz1 [13]. Consistent with previous Co-IP data [13,33,34], our biochemical analysis using purified proteins showed that a short and highly conserved fragment of Tpz1 (residues 185–212) is sufficient to maintain a stable interaction with $Pot1_{372-555}$ (S5A and S5B Fig). Hereafter, we refer to $Tpz1_{185-212}$ as the Pot1-interacting motif of Tpz1 ($Tpz1_{PIM}$) (Fig 1A). Multiple sequence alignment of Tpz1 proteins reveals that this region of Tpz1 is highly conserved in different fission yeast species (Figs 2A and S5C). We crystallized the $Pot1_{372-555}$-$Tpz1_{PIM}$ complex and determined its structure by single-wavelength anomalous dispersion (SAD) method at a resolution of 2.6 Å (Figs 2B and S5D and S1 Table).

The crystal structure reveals that $Pot1_{372-555}$ adopts a typical OB-fold architecture containing a highly curved five-stranded β-barrel, hereafter referred to as $Pot1_{OB3}$ (Figs 1A, 2B and S3). In the $Pot1_{OB3}$-$Tpz1_{PIM}$ complex, the $Tpz1_{PIM}$ polypeptide exhibits an extended conformation with three separated helices (H1, H2 and H3) (Fig 2B and 2C). Helices H1 and H2 of $Tpz1_{PIM}$ lie in a continuous groove on the concaved side of the $Pot1_{OB3}$, whereas helix H3 sits on a positively charged surface on the other side of $Pot1_{OB3}$ (Fig 2B and 2C). The formation of the binary complex buries a total of ~1,200 Å$^2$ solvent exposed surface area at the interface.

## The $Pot1_{OB3}$-$Tpz1_{PIM}$ interface

In the $Pot1_{OB3}$-$Tpz1_{PIM}$ complex structure, helices H1, H2 and H3 in $Tpz1_{PIM}$ divide the polypeptide into three adjacent binding modules for $Pot1_{OB3}$ (Fig 2B, 2C, 2D, 2E and 2F). The N-terminal $3_{10}$ helix H1 of $Tpz1_{PIM}$ and its peripheral residues fit into a hydrophobic depression formed by strands β1, β2 and β3 of $Pot1_{OB3}$ (Fig 2C and 2D). In particular, the side chain of Met190 and Leu193 in helix H1 points into two adjacent hydrophobic pockets formed by Trp417, Leu441 and Pro451, and by Trp417, Met437 and Ile453, respectively (Fig 2D). The short H2 helix fits into a narrow cleft formed by a two-stranded β1'- β2 protrusion and the long loop between strands β4 and β5 of $Pot1_{OB3}$, with the sidechain of Cys198 pointing into the deep hydrophobic groove (Fig 2B, 2C and 2E). This configuration is further stabilized by multiple electrostatic interactions between sidechains of Tpz1-Glu197 and Pot1-Arg546,

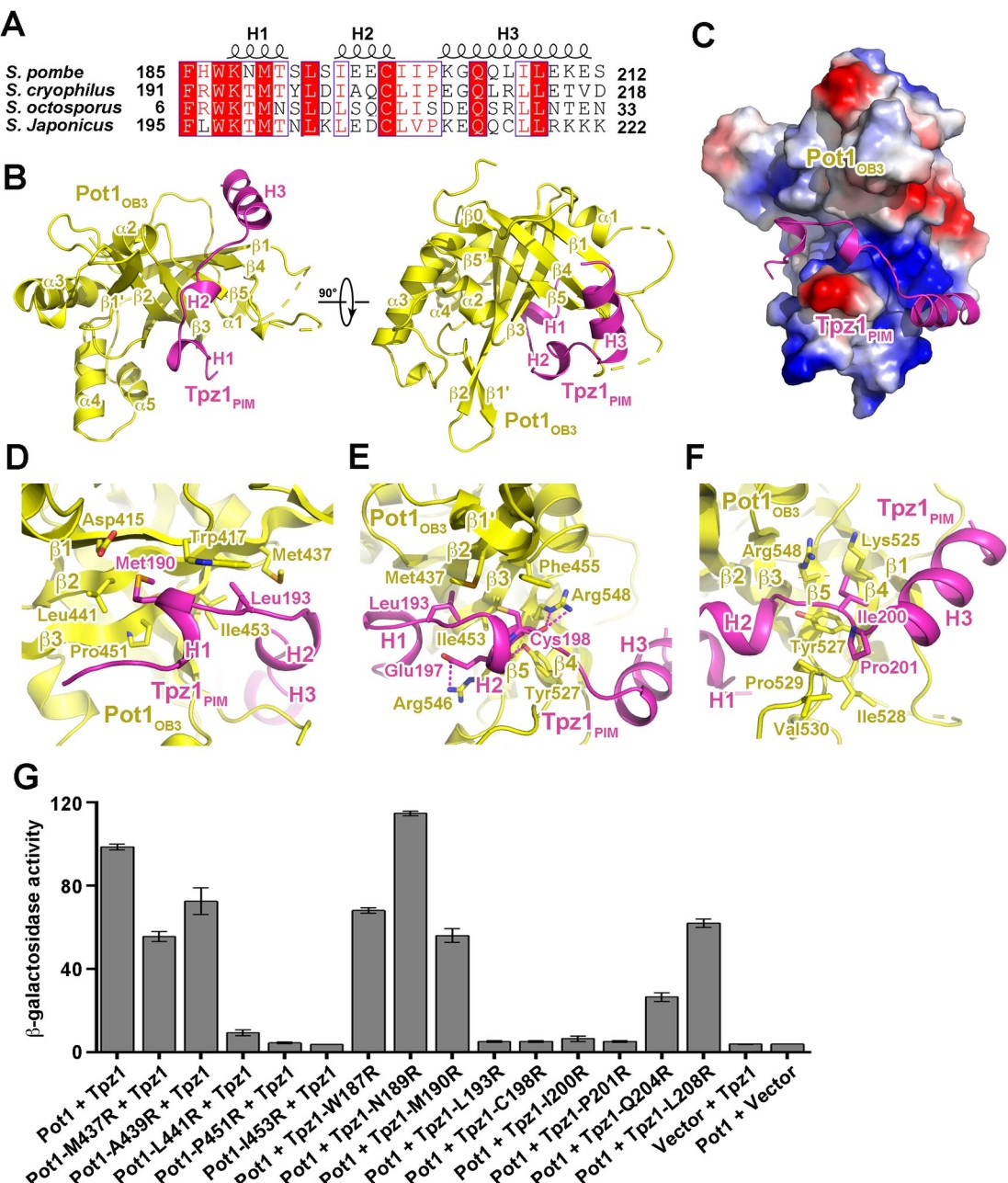

**Fig 2. Structural and mutational analyses of the Pot1$_{OB3}$-Tpz1$_{PIM}$ interaction.** (A) Multiple sequence alignment of *S. pombe* Tpz1$_{PIM}$ and its homologues. Conserved residues of Tpz1$_{PIM}$ are boxed and highlighted in red. (B) Overall structure of the Pot1$_{OB3}$-Tpz1$_{PIM}$ complex in two orthogonal views. Pot1$_{OB3}$ is colored in yellow, and Tpz1$_{PIM}$ in magenta, respectively. (C) Electrostatic surface potential of the Tpz1$_{PIM}$-binding site of Pot1$_{OB3}$. Positive potential, blue; negative potential, red. Tpz1$_{PIM}$ is shown in ribbon model and colored in magenta. (D) Details of the interactions between Tpz1$_{PIM}$ and Pot1$_{OB3}$ at the Tpz1$_{PIM}$ H1-binding interface. Residues that mediate the interactions at the interface are shown in stick model. (E) Details of the interactions between Tpz1$_{PIM}$ and Pot1$_{OB3}$ at the Tpz1$_{PIM}$ H2-binding interface. (F) Details of the interactions between Tpz1$_{PIM}$ and Pot1$_{OB3}$ at the Tpz1$_{PIM}$ H3-binding interface. (G) Effects of mutations on the Pot1$_{OB3}$-Tpz1$_{PIM}$ interaction were examined in yeast two-hybrid assays. Data are averages of three independent β-galactosidase measurements normalized to the wild-type interaction, arbitrarily set to 100. Error bars in the graph represent mean ± s.e.m. from three independent experiments.

between main chain carbonyls of Tpz1-Glu197 and Tpz1-Cys198 and sidechains of Pot1-Tyr527 and Pot1-Arg548, respectively (Fig 2E). In contrast to helices H1 and H2, helix H3 and its peripheral residues are away from the major binding groove of $Pot1_{OB3}$, attaching to a rather flat surface of $Pot1_{OB3}$ formed by strands β1, β4, and β5 through both electrostatic and hydrophobic contacts (Fig 2B, 2C and 2F). Between helices H2 and H3, Tpz1-Ile200 and Tpz1-Pro201 mediate van der Waals contacts with the aliphatic and aromatic sidechains of a panel of Pot1 residues, helping secure Tpz1 H3 helix on $Pot1_{OB3}$ (Fig 2F). In accordance with the crystal structure, previous mutagenesis data showed that mutations of key residues at the interface, Pot1-I453R and Tpz1-I200R, could completely disrupt the Pot1-Tpz1 interaction and leads to over-elongation of telomeres [17,33].

To further corroborate the structural analysis, we performed yeast two-hybrid (Y2H) experiments to validate the observed interactions between $Pot1_{OB3}$ and $Tpz1_{PIM}$. Consistent with the crystal structure, single amino-acid substitution of Pot1-Leu441, Pot1-pro451, Pot1-Ile453 or Tpz1-Leu193 at the H1-binding interface, or Tpz1-Cys198, Tpz1-Ile200 or Tpz1-Pro201 at the H2-binding interface with positively charged arginine residue completely abolished the interaction between $Pot1_{OB3}$ and $Tpz1_{PIM}$ (Fig 2G). Individual arginine substitution of Tpz1-Trp187, Tpz1-Asn189 and Tpz1-Met190 showed no effect on the $Pot1_{OB3}$-$Tpz1_{PIM}$ interaction (Fig 2G), suggestive of little contribution of the N-terminal of $Tpz1_{PIM}$ to $Pot1_{OB3}$ interaction. Notably, the Tpz1-Q204R mutation weakened, but did not disrupt the $Pot1_{OB3}$-$Tpz1_{PIM}$ interaction, consistent with the observation that Tpz1-Gln204 protrudes away from the major $Tpz1_{PIM}$-binding groove, and only contributes to two hydrogen-bonding interactions with $Pot1_{OB3}$ (Fig 2F and 2G). As a control, none of these mutations affected the interactions of Tpz1 with Ccq1 (S5E Fig).

## Structural conservation and divergence of *S. pombe* Pot1-Tpz1, human POT1-TPP1 and *O. nova* TEBPα-β complexes

Previous bioinformatics and structural studies have suggested a high structural similarity of *S. pombe* Pot1-Tpz1 to human POT1-TPP1 and *O. nova* TEBPα-β complex [35,36]. Consistently, the structure of $Pot1_{OB3}$ can be superimposed onto $TEBPα_{OB3}$ with an rmsd of 1.9 Å in the positions of 163 equivalent Cα atoms, and onto $POT1_{OB3}$ with an rmsd of 1.6 Å in the positions of 156 equivalent Cα atoms (S6A and S6B Fig). In all these OB folds, the β barrels form a canonical concaved groove that mediates the binding with their interacting partners Tpz1, TEBPβ and TPP1 (S6C, S6D and S6E Fig).

Despite these similarities, how the OB folds recognize their partners display some unique features in the three complexes. In the *S. pombe* $Pot1_{OB3}$-$Tpz1_{PIM}$ complex, the $Tpz1_{PIM}$ polypeptide fits into the binding groove of $Pot1_{OB3}$ and unidirectionally wraps around the OB fold (S6C Fig). In contrast in the *O. nova* TEBPα-β complex, in addition to the interaction as in the *S. pombe* $Pot1_{OB3}$-$Tpz1_{PIM}$ complex, TEBPβ has a longer C-terminal extension that makes a U-turn and folds back onto the surface of $TEBPα_{OB3}$ (S6D Fig) [36]. Human POT1-TPP1 interaction is the most divergent among the three complexes. POT1 has a Holliday Junction Resolvase-like (HJRL) domain inserted within the OB3 fold, so that $TPP1_{PIM}$ adopts an extended conformation to cover the surface of both OB3 and HJRL modules of POT1 (S6E Fig) [35,37]. These structural variations likely have evolved to meet the special functional need in different organisms.

## Crystal structure of the $Tpz1_{CBM}$-$Ccq1_{TAD}$ complex

In *S. pombe* shelterin complex, the Tpz1-Ccq1 interaction functions as a bridge between shelterin and protein complexes that coordinate telomere homeostasis and establish telomeric

heterochromatin structures [16,18,20]. To gain structural insights into Ccq1-dependent telomere functions, we set out to determine the crystal structure of the Tpz1-Ccq1 complex. Consistent with previous studies [16,17], we found that the N-terminal region of Ccq1 (residues 123–439) can stably interact with a short C-terminal fragment of Tpz1 (residues 425–470) (S7A Fig). Initial crystallization trials of the $Tpz1_{425-470}$-$Ccq1_{123-439}$ complex generated crystals that only diffracted to ~ 4.0 Å resolution. Multiple sequence alignment analysis of Ccq1 proteins from various species revealed several regions that are highly variable in sequence (S7B Fig). After an extensive screening of deletion mutations, we identified a construct of $Ccq1_{123-439}$ with a deletion of residues 199–215 ($Ccq1_{123-439\ \Delta199-215}$), which could form a stable binary complex with $Tpz1_{425-470}$ and produce high-quality crystals for structural studies. Hereafter, for simplicity, we refer to $Ccq1_{123-436\ \Delta199-215}$ and $Tpz1_{425-470}$ as the $Ccq1_{TAD}$ (Tpz1-associating domain) and $Tpz1_{CBM}$ (Ccq1-binding motif), respectively (Fig 1A). We determined the $Tpz1_{CBM}$-$Ccq1_{TAD}$ complex structure by the SAD method at a resolution of 2.4 Å (Fig 3A and S1 Table). The calculated electron density map allowed unambiguous tracing of most of the complex except for several disordered regions of $Ccq1_{TAD}$ (residues 123–131, 274–280 and 368–395) (Figs 3A and S8A).

The $Tpz1_{CBM}$-$Ccq1_{TAD}$ complex structure reveals a 2:2 stoichiometry and buries a total of ~2,100 $Å^2$ surface area in the complex (Fig 3A). Consistent with this observation, experiments using calibrated gel-filtration chromatography showed that the elution peak of the $Tpz1_{CBM}$-$Ccq1_{TAD}$ complex corresponds to a molecular weight of about 60 kDa (S7A Fig), as expected if the heterotetramer is present in solution. The $Tpz1_{CBM}$-$Ccq1_{TAD}$ complex exhibits a butterfly-shaped architecture with two $Ccq1_{TAD}$ molecules as the 'wings' and $Tpz1_{CBM}$ as the 'antenna' (Fig 3A). The two $Tpz1_{CBM}$ polypeptides adopt symmetric conformations and each $Tpz1_{CBM}$ contains an N-terminal loop and a C-terminal helix, which respectively interact with the two $Ccq1_{TBM}$ molecules in the heterotetramer complex (Fig 3A). Each $Ccq1_{TAD}$ folds into a globular domain, containing 11 helices and six β sheets (Fig 3A). Structural database search using the Dali server [38] revealed a structural resemblance of $Ccq1_{TAD}$ with the N-terminal domain of histone deacetylase (HDAC) complex subunit 3 (Hda3) from budding yeast *S. cerevisiae* (S8B Fig) [39], consistent with the previous bioinformatics prediction of $Ccq1_{TAD}$ as an HDAC2/3-like domain [18,40].

## The $Ccq1_{TAD}$ dimeric interface

In the center of the $Tpz1_{CBM}$-$Ccq1_{TAD}$ complex, the two $Ccq1_{TAD}$ molecules mediate a dimeric contact in a head-to-head fashion, burying ~500 $Å^2$ surface area between the two monomers (Fig 3A, 3B and 3C). The core of this symmetric dimer interface is mediated by helices α1 and α2 and the short loop between them from both $Ccq1_{TAD}$ subunits (Fig 3B and 3C). The most salient feature of this interface is the bipartite distribution of the electrostatic surface potential, positive at one end and negative at the other (Fig 3B). Such a configuration allows the two $Ccq1_{TAD}$ molecules to contact each other reciprocally in an energetically favorable manner (Fig 3B); the side chains of His161 and Arg170 of one $Ccq1_{TAD}$ respectively point into the basic and acidic depressions of the opposite $Ccq1_{TAD}$ subunit, coordinating a total of six electrostatic interactions (Fig 3C). Although the dimeric $Ccq1_{TAD}$ interface is predominantly hydrophilic, intermolecular hydrophobic interactions provide additional specificity and stability to the dimer. Two pairs of symmetry-related residues, Phe157 and His161 from helix α2 and Ile171 from α3 in both $Ccq1_{TAD}$ molecules, mediate intimate hydrophobic contacts between the two monomers at the center of the interface (Fig 3C). Although this intimate interface is not enough for the dimer formation of $Ccq1_{TAD}$ by itself (Fig 3D) [41], the close vicinity of this interface to the $Tpz1_{CBM}$-$Ccq1_{TAD}$ contacts indicate that the dimeric

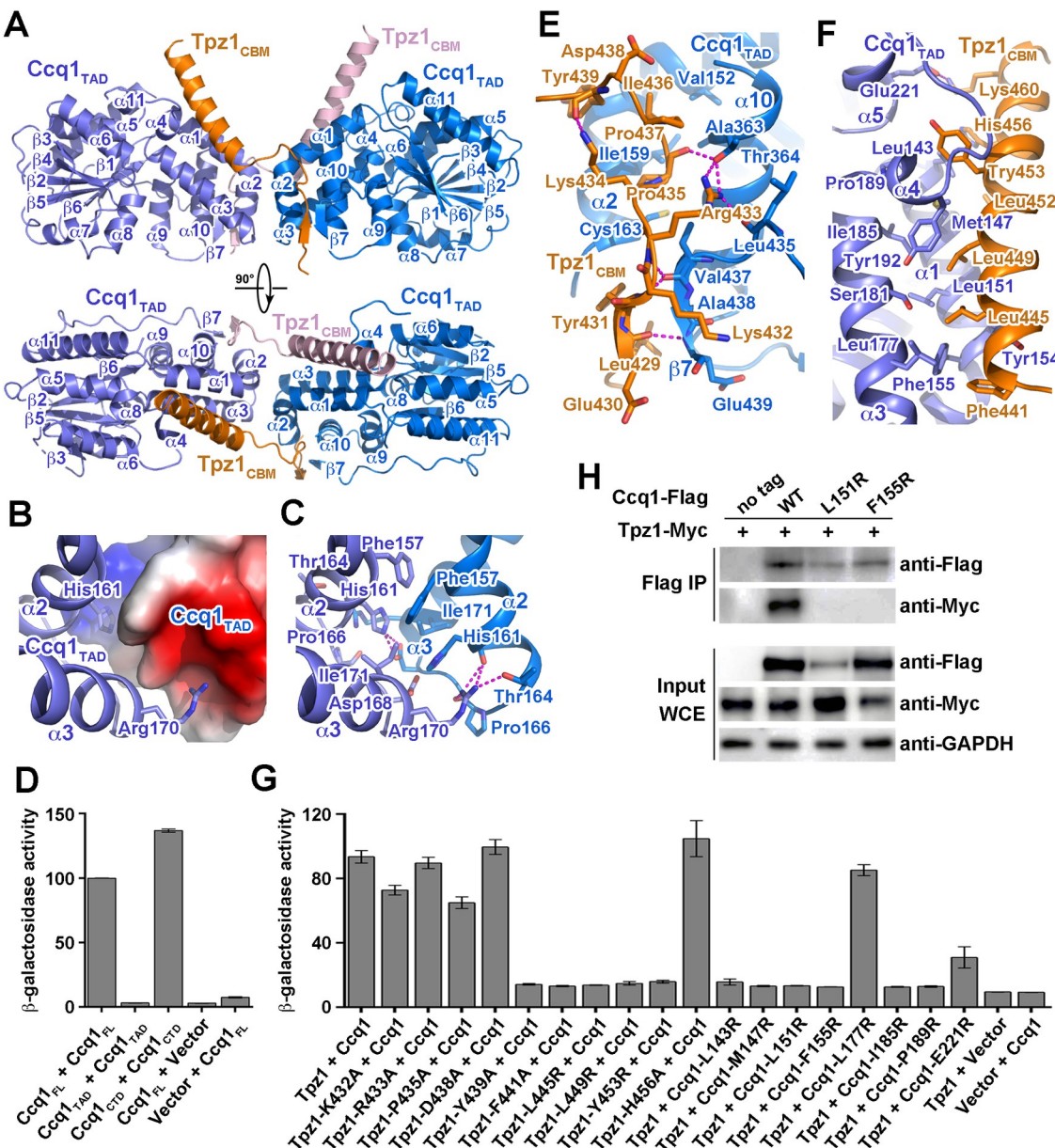

**Fig 3. Structural and mutational analyses of the Tpz1$_{CBM}$-Ccq1$_{TAD}$ interaction.** (A) Overall structure of the Tpz1$_{CBM}$-Ccq1$_{TAD}$ complex in two orthogonal views. Ccq1$_{TAD}$ is colored in slate blue and marine blue, and Tpz1$_{CBM}$ in orange and pink, respectively. (B) Electrostatic surface potential of the Ccq1$_{TAD}$ dimeric interface. Positive potential, blue; negative potential, red. One Ccq1$_{TAD}$ is presented in ribbon model and colored in slate blue. (C) Details of the intermolecular interactions at the Ccq1$_{TAD}$ dimeric interface. Residues important for the interactions are shown in stick model and hydrogen bonds are denoted as magenta dashed lines. (D) Identification of regions of Ccq1 that mediate dimer formation. Data are averages of three independent β-galactosidase measurements normalized to the full-length Ccq1 dimer interaction, arbitrarily set to 100. Error bars in the graph represent mean ± s. e.m. from three independent experiments. (E) Details of the interactions between Tpz1$_{CBM}$ and Ccq1$_{TAD}$ at the N-terminal loop of Tpz1$_{CBM}$. Residues important for the interactions are shown in stick model and hydrogen bonds are denoted as magenta dashed lines. (F) Details of the interactions between Tpz1$_{CBM}$ and Ccq1$_{TAD}$ at the C-terminal helix of Tpz1$_{CBM}$. Residues that mediate the interactions are shown in stick model. (G) Effects of mutations at the Tpz1$_{CBM}$-Ccq1$_{TAD}$ interface on the Tpz1-Ccq1 interaction were examined in yeast two-hybrid assays. Data are averages of three independent β-galactosidase measurements normalized to the wild-type interaction, arbitrarily set to 100. Error bars in the graph represent mean ± s.e.m. from three independent experiments. (H) Co-IP analysis of the interaction of Myc-tagged Tpz1 with Flag-tagged wild-type or mutant Ccq1.

conformation of Ccq1$_{TAD}$ is a prerequisite for the stable interaction between Ccq1$_{TAD}$ and Tpz1$_{CBM}$. Indeed, previous mutagenesis data showed that mutations of two important residues in the dimeric interface (Ccq1-I171R and Ccq1-F157A) could impair the Tpz1-Ccq1 interaction [33], suggesting that the dimeric interface of Ccq1$_{TAD}$ is essential for the heterotetrameric architecture of the Tpz1$_{CBM}$-Ccq1$_{TAD}$ complex. Notably, the C-terminal domain of Ccq1 (Ccq1$_{CTD}$) alone can mediate Ccq1 homodimerization (Fig 3D) [42]. However, it was reported that the Ccq1-Tpz1-Poz1 complex dimerizes in the absence of the Ccq1$_{CTD}$ [42], reminiscent of the heterodimeric structure of the Tpz1$_{CBM}$-Ccq1$_{TAD}$ complex (Fig 3A, 3B and 3C). Thus, we conclude that the Ccq1$_{TAD}$ dimeric interface in the Tpz1$_{CBM}$-Ccq1$_{TAD}$ structure and the Ccq1$_{CTD}$ homodimer together promote the heterodimerization of the Tpz1-Ccq1 complex.

## The Tpz1$_{CBM}$-Ccq1$_{TAD}$ interface

In the Tpz1$_{CBM}$-Ccq1$_{TAD}$ complex, the N-terminal loop of Tpz1$_{CBM}$ adopts an extended conformation, meandering in a shallow, acidic groove formed by helices α2 and α10 as well as the C-terminal end of one Ccq1$_{TAD}$ molecule, primarily stabilized by a panel of electrostatic and hydrogen-bonding interactions (Figs 3A, 3E, and S8C). Notably, the N-terminal residues $_{430}$EYK$_{432}$ of Tpz1$_{CBM}$ form a short intermolecular β sheet with the C-terminal residues $_{437}$VAE$_{439}$ of Ccq1$_{TAD}$ (Fig 3A and 3E). The surfaces of the Tpz1$_{CBM}$ loop and the Ccq1$_{TAD}$ groove are not only opposite in charge distribution but also complementary in shape (Figs 3E and S8C). While electrostatic interactions should favor the initial apposition of the two proteins, the interaction specificity between Tpz1$_{CBM}$ and Ccq1$_{TAD}$ is mainly provided by van der Waals contacts (Fig 3E). The hydrophobic side chains of Tpz1$_{CBM}$ Leu429, Tyr431, Pro435 and Ile436 point into two hydrophobic depressions along the Ccq1$_{TAD}$ groove, accounting for about half of the total buried surface area between Tpz1$_{CBM}$ and Ccq1$_{TAD}$ (Fig 3E). Consistently, previous mutagenesis data showed that two mutations (Ccq1-C163R and Ccq1-A363R) in this hydrophobic interface could disrupt the Tpz1-Ccq1 interaction [18]. The Tpz1$_{CBM}$ polypeptide makes a sharp turn at Asp438 and Tyr439 so that the C-terminal helix makes direct contacts with the other Ccq1$_{TAD}$ monomer in the complex (Fig 3A and 3E). The hydrophobic portion of the amphipathic helix of Tpz1$_{CBM}$ fits into a long groove formed by helices α1, α3 and α4 of Ccq1$_{TAD}$ through extensive hydrophobic contacts (Fig 3F). Two point mutations, Tpz1-L449R and Ccq1-L151R, were reported to abolish the interaction [16,17,33], consistent with the structure that the side chain of Tpz1-Leu449 points into a hydrophobic depression surrounded by Ccq1 residues Met147, Arg150, Leu151, Ile185, and Tyr192 (Fig 3F).

To further examine the significance of the Tpz1$_{CBM}$-Ccq1$_{TAD}$ heterotetrameric interface, we assessed the effects of an additional panel of mutations in either Tpz1$_{CBM}$ or Ccq1$_{TAD}$ using Y2H analysis. Mutations of the N-terminal loop of Tpz1$_{CBM}$ showed little effect on the Tpz1-Ccq1 interaction (Fig 3G). However, Tpz1$_{CBM}$ mutations of either hydrophobic residue in the middle of the amphipathic helix (Y453R) or residues in the junction region between the loop and the helix (F441R and Y439R) were sufficient to abolish the Tpz1-Ccq1 interaction (Fig 3G). As a control, none of these mutations affected the interactions of Tpz1 with Pot1 (S8D Fig). Similarly, mutations of Ccq1 hydrophobic residues on the other side of the interface (L143R, M147R, F155R, I185R, and P189R) can completely disrupt its interaction with Tpz1 (Fig 3G). In addition, co-immunoprecipitation analysis showed that the Tpz1-Ccq1 interaction was completely disrupted by the Ccq1-L151R and Ccq1-F155R mutations in yeast cells (Fig 3H). Notably, although Ccq1-L151R protein exhibited reduced expression, Ccq1-F155R protein maintained the WT Ccq1 expression (Fig 3H), suggestive of the correct folding of Ccq1-F155R mutant protein that only disrupt the interaction with Tpz1. Collectively, we

conclude that hydrophobic contacts at the interface are the major driving force underlying the Ccq1-Tpz1 interaction.

## The Tpz1-Ccq1 interaction is essential for telomere maintenance and telomeric heterochromatin formation

To investigate the *in vivo* function of the Tpz1-Ccq1 interaction, we generated a panel of yeast strains with Tpz1-binding-deficient mutations in Ccq1 (*ccq1-L143R*, *ccq1-L151R*, *ccq1-F155R* and *ccq1-P189R*), and examined their effects on telomere length maintenance and telomeric hetero-chromatin formation. Southern blotting analysis revealed that these strains exhibited telomere shortening (Fig 4A), reminiscent of progressive telomere shortening and HR-dependent telomere maintenance in *ccq1Δ* cells [40,43]. These results support the notion that disruption of the Tpz1-Ccq1 interaction impairs the Ccq1-dependent telomerase recruitment [16,17]. Furthermore, we also tested the effect of these mutants on heterochromatin formation at telomeres by examining transcriptional silencing of a *his3*+ reporter gene inserted adjacent to telomere IL. Our results showed that these mutant strains failed to transcriptionally repress the *his3*+ expression (Fig 4B and 4C and S2 Table), suggesting that disruption of the Tpz1-Ccq1 interaction leads to a defect in telomeric heterochromatin formation [18]. As a negative control, the Ccq1-L177R mutation, that could not disrupt the Tpz1-Ccq1 interaction in Y2H assay, exhibited no effect on telomere length maintenance and telomeric heterochromatin formation (S9A and S9B Fig). To further understand the impact of the Tpz1-Ccq1 interaction on telomeric heterochromatin formation, we investigated the expressions of TERRA (telomeric repeat-containing non-coding RNA) from telomere adjacent regions and the *tlh1*+ gene 15 kb away from the telomeric repeats by reverse transcription-quanti-tative PCR (RT-qPCR). The result clearly showed that transcripts of both TERRA and the *tlh1*+ gene were highly increased in *ccq1-F155R* compared with wild-type (WT) cells (Fig 4C and S2 Table). As a control, transcription of centromeric region (*cen-dg*) was unaffected in *ccq1-F155R* cells (Fig 4C and S2 Table). Together, these results reveal the essential role of the Tpz1-Ccq1 inter-action in telomere maintenance and telomeric heterochromatin formation.

To further understand the *in vivo* function of the Tpz1-Ccq1-interaction, we tested whether the Tpz1-binding deficient Ccq1 mutants affected the telomere association of Ccq1. Chroma-tin immunoprecipitation (ChIP) data revealed that Ccq1-F155R mutations led to a great elimi-nation of Ccq1 from telomeres (Figs 4D and S9C and S2 Table). Because Ccq1-F155R protein maintained the WT Ccq1 expression (Fig 3H), the loss of the Tpz1-Ccq1 interaction was mainly responsible for the Ccq1 elimination in the *ccq1-F155R* cells. Notably, the amounts of telomeres associated Tpz1 and Rap1 were also decreased in the *ccq1-F155R* cells (Figs 4E, 4F, S9D and S2 Table), likely due to the telomere shorting after the disruption of the Tpz1-Ccq1 interaction (Fig 4A). Our ChIP data showed that telomeric association of telomerase catalytic subunit Trt1 was greatly decreased in the *ccq1-F155R* cells (Figs 4G and S9D and S2 Table). Moreover, RNA-immunoprecipitation (RIP) data showed that the Ccq1-F155R mutation failed to coimmunoprecipitate with telomerase RNA *TER1* (S9E Fig and S2 Table). These results are in line with previous reports that the Tpz1-Ccq1 interaction-deficient mutants affect the recruitment of telomerase [16]. Furthermore, ChIP data also revealed that telomeric enrichments of Clr3 (the deacetylase subunit of SHREC) and Clr4 (the methyltransferase sub-unit of CLRC) were markedly reduced in the *ccq1-F155R* cells (Figs 4H, 4I, S9D and S2 Table). Taken together, we conclude that the Tpz1-Ccq1 interaction plays an essential role in Ccq1 recruitment to telomeres that functions as a platform for telomerase and heterochromatic complexes SHREC and CLRC, reinforcing the notion that the Tpz1-Ccq1 subcomplex func-tions as a molecular bridge to coordinate telomere homeostasis and to establish the hetero-chromatin structure at telomeres [16–18].

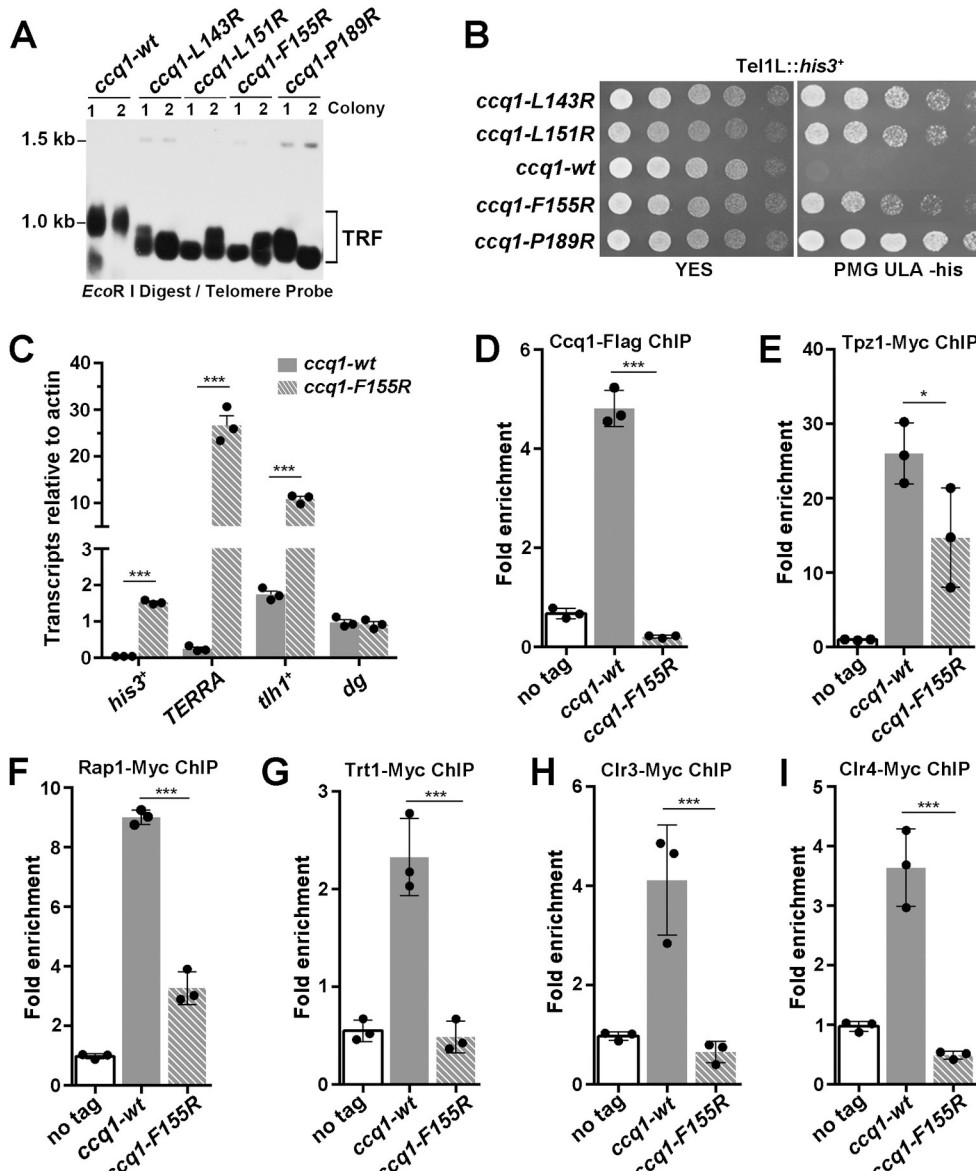

**Fig 4. Functional analysis of the Tpz1-Ccq1 interaction in telomere maintenance and telomeric heterochromatin formation.** (A) Southern blot analysis of telomere lengths of wild-type or Tpz1-binding deficient Ccq1 mutant strains. Genomic DNAs were digested with *Eco*R I and subjected to Southern blot analysis with a telomere-specific probe. (B) Effects of Tpz1-binding deficient mutations of Ccq1 on the transcriptional silencing of *his3*+ reporter gene inserted adjacent to the telomeric region. Equal amounts of 10-fold dilution series of cultures were spotted on YES or Pombe Medium Glutamate supplemented with uracil, leucine, and adenine (PMG ULA) (-histidine) plates. (C) RT-qPCR analysis of the transcription of *TERRA* and *tlh1*+ in the wild-type and *ccq1-F155R* cells. The transcription at the *cen-dg* region was used as a control. Data are represented as mean ± s.e.m. from three independent experiments. (D-I) Effects of the Ccq1-F155R mutation on telomere association for Ccq1 (D), Tpz1 (E), Rap1 (F), the telomerase catalytic subunit Trt1 (G), the deacetylase subunit Clr3 of the SHREC complex (H) and the methyltransferase subunit Clr4 of the CLRC complex (I) were measured by ChIP-qPCR assays. Recruitment to the internal *act1*+ locus serves as a control for ChIP specificity. Data are represented as mean ± s.e.m. from three independent experiments.

## Discussion

The multi-subunit and highly flexible nature of *S. pombe* shelterin complex has greatly impeded our structural and functional understanding for this important complex in fission

yeast. In previous studies, we have determined the crystal structures of the Poz1-Tpz1-Rap1, Taz1-Rap1 subcomplexes and various domains of Taz1 and Pot1 [14,15,21,26,44]. Here, we report the crystal structures of three additional modules, Pot1-ssDNA, Pot1-Tpz1 and Ccq1-Tpz1, of the shelterin complex, which enables us to build an atomic model for the entire *S. pombe* shelterin complex (S10 Fig), when integrated with the three-dimensional arrangement of the CTP complex [42]. In this model, the close proximity between the Poz1- and Ccq1-interacting motifs in Tpz1 imposes a strong spatial constraint on the arrangement between the Poz1-Tpz1 and the Ccq1-Tpz1 modules so that Poz1, Tpz1 and Ccq1 together form a $(Poz1-Tpz1-Ccq1)_2$ heterohexameric central hub in the shelterin complex between the ds and ss regions of the telomere (S10 Fig). From this hub extend out two copies of the Taz1-Rap1 and Pot1-Tpz1 modules that bind to dsDNA and ssDNA regions, respectively (S10 Fig). Among the six subunits of the shelterin complex, Taz1, Poz1, and the Ccq1-Tpz1 subcomplex are intrinsically dimer by themselves (Fig 3) [14,15,45], defining the overall dimeric conformation of the shelterin complex (S10 Fig).

The shelterin complex is a conserved telomeric DNA-binding complex from fission yeast to mammals. To our knowledge, so far, all the domains and subcomplexes in the either *S. pombe* or human shelterin complex have been structurally characterized [14,15,21, 24,26,35,37,44,46,47], allowing us to compare the similarities and differences between *S. pombe* and human shelterin complexes (Fig 5A). The most conserved structural feature between the two shelterin complexes is the overall bridge conformation linking ds and ss regions of telomeres, although the shelterin architecture and organization display some unique features in *S. pombe* and human (Fig 5A). In contrast with dimerization of only dsDNA binders TRF1 and TRF2 in human shelterin [48,49], Taz1, Poz1, and the Ccq1-Tpz1 subcomplex form homodimers by themselves that defines the overall dimeric conformation of the *S. pombe* shelterin complex (Figs 3 and 5A) [14,15,44]. The ssDNA-binding Pot1/POT1 is highly conserved, both of which harbor a N-terminal dual OB folds (DBD) for the ssDNA-binding activity and a C-terminal OB fold (OB3) for interaction with Tpz1/TPP1. However, the human $POT1_{DBD}$ function together as a single entity to recognize a regular human telomeric sequence [24]; while, the $Pot1_{OB1}$ and $Pot1_{OB2}$ in *S. pombe* $Pot1_{DBD}$ are structurally separable, which together with the long flexible loop between them recognize degenerate telomeric sequences in *S. pombe* (Figs 1 and S3). The human POT1-TPP1 interaction is much more extensive owing to the insertion of a HJRL domain in the $POT1_{OB3}$, comparable with that of *S. pombe* Pot1-Tpz1 complex (Figs 2 and S6) [35]. The Pot1-Tpz1/POT1-TPP1 complex is a telomerase recruiter in both *S. pombe* and human shelterin complexes. The human POT1-TPP1 complex recruits telomerase via an TPP1-TERT interaction, and serves as a processivity factor for telomerase [22,50,51]. While, the *S. pombe* Pot1-Tpz1 complex recruits telomerase through the Tpz1-Ccq1-Est1 interaction network [16,17]. Moreover, the Tpz1-Ccq1 interaction also serves as platform for heterochromatic complexes [18,20]. It is likely that the shelterin complex has evolved distinct molecular architectures to accommodate different functions in fission yeast and mammals during evolution.

Our structural and functional data reported here, when combined with previous studies, provide an integrated picture for telomere maintenance, telomere protection and telomeric heterochromatin formation in fission yeast (Fig 5B). *S. pombe* utilizes the flexibly tethered dual OB folds of Pot1 to accommodate heterogeneous telomeric DNA (S3 Fig) [19,30,32,52], and the Tpz1-Pot1 interaction to link Pot1 to the telomeric dsDNA that avoids Pot1 binding to non-telomeric regions (Fig 2) [13,22]. It should be noted that $Pot1_{DBD}$ is able to bind telomeric ssDNA at least in two modes that modulates 3' end accessibility for telomerase [52]. In the late S phase, the Pot1 binds telomeric ssDNA in the manner with the 3' end of ssDNA unbound by $Pot1_{OB2}$ and accessible to telomerase; meanwhile, the Pot1-Tpz1 complex recruits telomerase

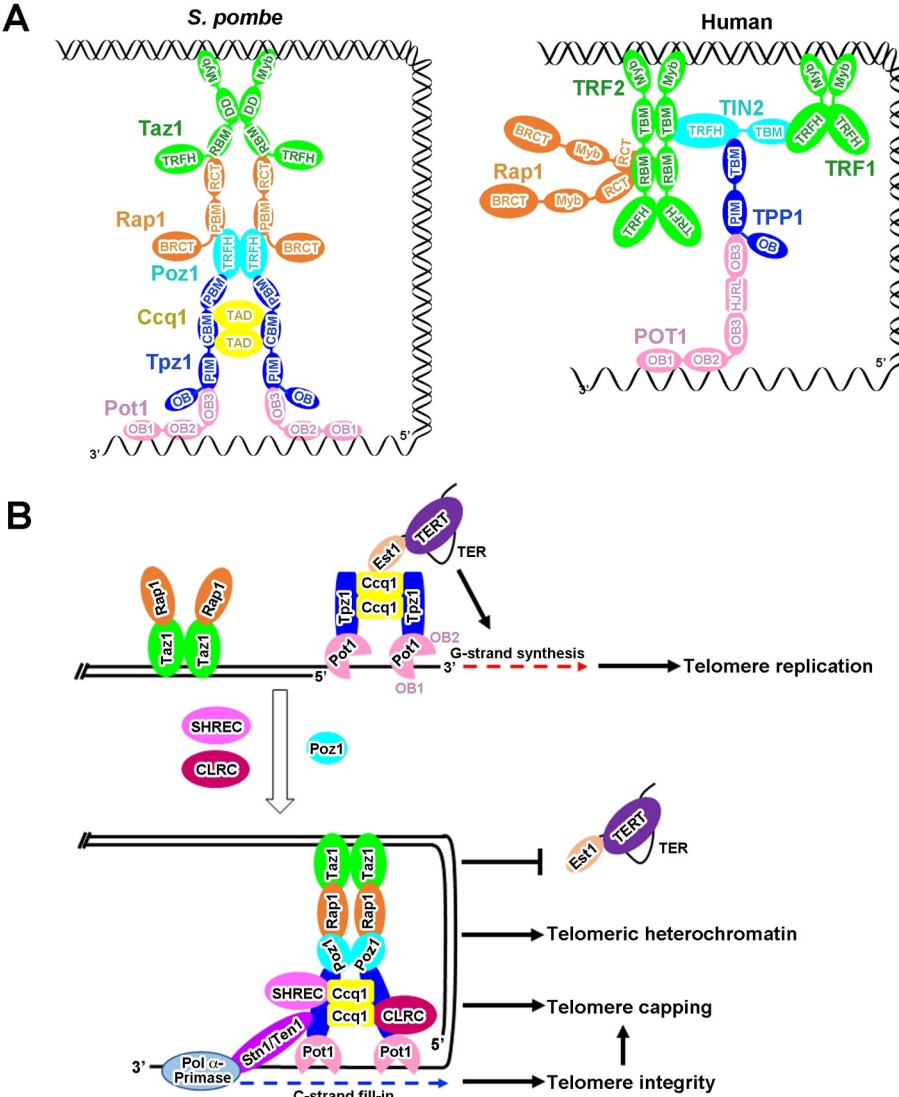

**Fig 5. The architecture of the *S. pombe* shelterin complex and a schematic model for its roles at telomeres.** (A) Modularized organization model for the *S. pombe* (left) and human (right) shelterin complexes based on available crystal structures of domains and subcomplexes in these two complexes. (B) Schematic model for telomere maintenance, telomere protection and telomeric heterochromatin formation by the shelterin complex in fission yeast *S. pombe*. When the cell cycle progresses into late S phase, the Tpz1-Ccq1-Est1 interaction network promotes telomerase recruitment for telomere extension. After telomeric G-strand synthesis, SHREC- and CLRC-associated Ccq1 dissociates telomerase from Tpz1 to prevent telomere over-elongation and to establish telomeric heterochromatin. Meanwhile, Poz1 interacts with both Rap1 and Tpz1 that form a bridge in the shelterin complex linking the dsDNA and ssDNA regions to recap telomere ends for telomere protection. Finally, the Stn1-Ten1 complex is recruited to telomeres via the interaction with Tpz1 to recruit the DNA Polα-primase complex for the C-strand fill-in synthesis to ensure telomere integrity.

for telomere extension through the Tpz1-Ccq1-Est1 interaction network [16,17,43,52]. After telomeric G-strand replication, Ccq1 recruits the SHREC and the CLRC complexes to telomeres to establish telomeric heterochromatin, and at the same time dissociates telomerase from telomeres to prevent telomere over-elongation [18,20,41]. Consistent with this idea, disruption of the Tpz1-Ccq1 interaction leads to reduced telomere association of both telomerase and heterochromatic complexes, causing defects in telomere maintenance and telomeric

heterochromatin formation (Fig 4) [16–18]. After telomerase is released from telomeres, the Stn1-Ten1 complex is recruited to telomeres via SUMOylation-mediated interaction with Tpz1 to recruit the Polα-primase complex for the C-strand fill-in synthesis [53,54]. Concordant with these events, Pot1 binds telomeric ssDNA in a non-extendible state, and the shelterin bridge Tpz1-Poz1-Rap1 assembles in a hierarchical manner between the telomeric dsDNA and ssDNA regions, transforming the telomere into a capping structure for telomeres protection [14,15,52].

In our model, the Tpz1-mediated central hub in the shelterin complex plays a key role in regulating telomeric homeostasis, end protection and heterochromatin formation (Fig 5B). An outstanding question is how Tpz1 regulates telomere homeostasis through its interactions with both positive regulator Ccq1 and negative regulator Poz1 of telomerase-dependent G-strand synthesis. In addition, it is still not clear how Polα-primase complex-mediated C-strand fill-in are coordinated with the establishment of heterochromatin at telomeres. We propose that these processes likely are coupled together through conformational changes induced by the interactions between Poz1-Tpz1-Ccq1 and different complexes in a highly orchestrated manner. Future studies will be required to fully understand how the *S. pombe* shelterin complex fulfill its essential functions in telomere protection, homeostasis regulation and telomeric heterochromatin formation.

## Materials and methods

### Protein expression and purification

$Pot1_{DBD}$ (residues 2–339), $Pot1_{OB3}$ (residues 357–555) and $Ccq1_{TAD}$ (123–439, with residues 199–215 deletion) were respectively cloned into a modified pET28a vector with a SUMO protein fused at the N terminus after the 6×His tag. $Tpz1_{PIM}$ (residues 164–240) and $Tpz1_{CBM}$ (residues 425–470) were respectively cloned into a modified pGEX vector with a GST tag. The $Pot1_{DBD}$ protein was expressed in *E. coli* BL21 (DE3) CodonPlus cells (Stratagene) [22]. For preparation of the $Pot1_{OB3}$-$Tpz1_{PIM}$ and $Tpz1_{CBM}$-$Ccq1_{TAD}$ complexes, the corresponding plasmids were co-expressed in *E. coli* BL21 (DE3) CodonPlus cells. After induction for 20 h with 0.2 mM IPTG at 20˚C, the cells were harvested by centrifugation and the pellets were resuspended in lysis buffer (50 mM Tris-HCl, pH 8.0, 500 mM NaCl, 10% glycerol, 1 mM PMSF, 5 mM benzamidine, 1 mg mL$^{-1}$ leupeptin and 1 mg mL$^{-1}$ pepstatin). The cells were then lysed by sonication and the cell debris was removed by ultracentrifugation. The supernatant was mixed with Ni-NTA agarose beads (QIAGEN) and rocked for 0.5 hours at 4˚C before elution with 250 mM imidazole. The ULP1 protease was added to remove the His-SUMO tag. The protein sample was then purified with glutathione Sepharose-4B beads (GE Healthcare) and rocked overnight at 4˚C before elution with 15 mM reduced glutathione (Sigma). PreScission protease was then added to remove the N-terminal GST tags. The proteins were further purified by Hitrap-Q and gel-filtration chromatography equilibrated with 25 mM Tris-HCl pH 8.0, 150 mM NaCl, and 5 mM dithiothreitol. The final purified proteins were concentrated to 15 mg mL$^{-1}$ and stored at −80˚C.

For preparation of the $Pot1_{DBD}$-Tel18 complex, purified $Pot1_{DBD}$ protein was incubated with ssDNA (Tel18) in a molecular ratio of 1:1.3 and further purified by Hitrap-Q and gel-filtration chromatography equilibrated with 25 mM Tris-HCl pH 8.0, 150 mM NaCl, and 5 mM dithiothreitol. Finally, the $Pot1_{DBD}$-Tel18 complex was concentrated to 15 mg mL$^{-1}$ and stored at −80˚C.

### Crystallization, data collection, and structure determination

Crystals of the $Pot1_{DBD}$-Tel18 complex were grown by sitting-drop vapor diffusion at 4˚C. The precipitant well solution consisted of 20% PEG3350, 100 mM ammonium sulfate and 100 mM

Bis-Tris pH 5.5. Crystals were gradually transferred into a harvesting solution containing 20% PEG3350, 100 mM ammonium sulfate, 100 mM Bis-Tris pH 5.5 and 25% glycerol, followed by flash-freezing in liquid nitrogen for storage. All the datasets were collected under cryogenic conditions (100K) at SSRF beamlines BL18U1 and BL19U1. A 3.0-Å native dataset of the $Pot1_{DBD}$-Tel18 complex was collected and the complex structure was solved by molecular replacement with searching models 1QZG and 4HIK. The model was then refined using Phenix [55], together with manual building in Coot [56]. In the final Ramachandran plot, the favored and allowed residues are 93.7% and 100.0%, respectively.

Crystals of the $Pot1_{OB3}$-$Tpz1_{PIM}$ complex were grown by sitting-drop vapor diffusion at 4˚C. The precipitant well solution consisted of 25% PEG8000, 100 mM sodium citrate and 100 mM Tris-HCl pH 8.0. Crystals were gradually transferred into a harvesting solution containing 25% PEG8000, 100 mM sodium citrate, 100 mM Tris-HCl pH 8.0 and 25% glycerol, followed by flash-freezing in liquid nitrogen for storage. Crystals of SeMet-labeled $Pot1_{OB3}$-$Tpz1_{PIM}$ complex were grown in the similar condition. All the datasets were collected under cryogenic conditions (100K) at SSRF beamlines BL18U1 and BL19U1. A 2.9-Å SeMet-SAD dataset of the $Pot1_{OB3}$-$Tpz1_{PIM}$ complex was collected at the Se peak wavelength (0.97853 Å) and was processed by HKL3000 [57]. Six selenium atoms were located and refined, and the initial SAD electron density map was calculated using Phenix [55]. The initial SAD map was substantially improved by solvent flattening. The model was then refined against a native dataset with 2.6-Å resolution using Phenix [55], together with manual building in Coot [56]. In the final Ramachandran plot, the favored and allowed residues are 97.4% and 100.0%, respectively.

Crystals of the $Tpz1_{CBM}$-$Ccq1_{TAD}$ complex were grown by sitting-drop vapor diffusion at 4˚C. The precipitant well solution consisted of 15% PEG4000, 200mM potassium chloride, 50 mM magnesium chloride and 50 mM Tris-HCl, pH 7.8. Crystals were gradually transferred into a harvesting solution containing 15% PEG4000, 200mM potassium chloride, 50 mM magnesium chloride, 50 mM Tris-HCl, pH 7.8 and 25% glycerol, followed by flash-freezing in liquid nitrogen for storage. Crystals of SeMet-labeled $Tpz1_{CBM}$-$Ccq1_{TAD}$ complex were grown in the similar condition. All the datasets were collected under cryogenic conditions (100K) at SSRF beamlines BL18U1 and BL19U1. A 2.8-Å SeMet-SAD dataset of the $Tpz1_{CBM}$-$Ccq1_{TBM}$ complex was collected at the Se peak wavelength (0.97853 Å) and was processed by HKL3000 [57]. Six selenium atoms were located and refined, and the initial SAD electron density map was calculated using Phenix [55]. The initial SAD map was substantially improved by solvent flattening. The model was then refined against a native dataset with 2.4-Å resolution using Phenix [55], together with manual building in Coot [56]. In the final Ramachandran plot, the favored and allowed residues are 96.8% and 100.0%, respectively.

All of the crystal data collection and refinement statistics were summarized in S1 Table, and all of the crystal structural figures were generated using PyMOL software (Schrodinger, LLC).

## Isothermal titration calorimetry

The equilibrium dissociation constants of $Pot1_{DBD}$-ssDNA interactions were determined using a MicroCal iTC200 calorimeter (Malvern). The binding enthalpies were measured at 20˚C in 25 mM Tris-HCl, pH 8.0 and 150 mM NaCl. ITC data were subsequently analyzed and fitted using Origin 7 software (OriginLab).

## Yeast two-hybrid assay

The yeast two-hybrid assay was performed as described previously [58]. Briefly, the L40 strain was transformed with pBTM116 and pACT2 (Clontech) fusion plasmids, and colonies

harboring both plasmids were selected on YC (Yeast complete)–Leu–Trp plates. The β-Galactosidase activities were measured by a liquid assay.

## Strains, gene tagging and mutagenesis

The growth media and basic genetic techniques were performed as previously described [59,60]. The yeast strain TN9125, carrying an integrated $his3^+$ marker adjacent to telomeric repeats of the chromosome IL, was a gift from Dr. Toru M. Nakamura [61]. Genes tagged with the 13×Myc or 3×FLAG epitope was introduced as described [62]. Briefly, a tag with hygMX6 ($hyg^r$) cassettes was inserted at the C-terminal of interesting genes by homologous recombination [62]. Mutations in the $ccq1^+$ gene with kanMX6 ($kan^r$) were created by PCR, and each mutated DNA fragment was integrated at the endogenous gene's locus. All strains used in this study are listed in S3 Table.

## Yeast growth on plates

Single colonies were inoculated into 5 ml of yeast extract with supplement (YES) and cultured to saturation. The cultures were then diluted to $OD_{600} = 1$, and equal amounts (5 μl) of tenfold serial dilutions of the cultures were spotted on YES or Pombe Medium Glutamate supplemented with uracil, leucine, and adenine (PMG ULA) (−histidine) plates. After incubation at 30˚C for 2–3 days, plates were photographed.

## Telomere southern blot

Telomere blot was performed as described previously [16–18]. Briefly, $ccq1$ mutant transformants were confirmed by PCR and sequencing. The cells were harvested from 5 ml liquid culture inoculated from YES plates. Genomic DNA was purified by using phenol chloroform method, digested with $Eco$R I, and fractionated by electrophoresis on 1.0% agarose gel. The DNA fragments were transferred to a Hybond-N$^+$ Nylon membrane (GE Healthcare), UV cross-linked and incubated with Church buffer for 30 min at 50˚C. Biotinylated telomeric-specific probe was incubated with the DNA at 50˚C overnight, and biotin-probe-bound DNA fragments corresponding to telomeric DNAs were detected using Chemiluminescent Nucleic Acid Detection Module (Thermo Scientific, USA).

## RT-qPCR analysis

Total RNA was isolated using RNeasy mini kit (QIAGEN). One microgram RNA was used as template for the reverse transcription of 20 μl cDNA using PrimeScript RT reagent Kit with gDNA Eraser (Perfect Real Time) (TAKARA). Two microliters of the RT reaction were used to analyze gene expression level by quantitative real-time polymerase chain reaction (PCR) and normalized to that of $act1^+$. The real-time PCR was performed in the LightCycler 480 (Roche), and the TB Green *Premix Ex Taq* II (Tli RNaseH Plus) (TAKARA) reagent was used. The qPCR conditions were 30 s at 95˚C, 40 cycles of 5 s at 95˚C for denaturation, 30 s at 60˚C for annealing and extension. The primers were used as described previously (S4 Table) [63].

## Co-immunoprecipitation (co-IP) and western blot analysis

Co-IP experiments were performed as described previously [16]. Whole-cell extracts were prepared in lysis buffer (50 mM HEPES, pH 7.5, 150 mM NaCl, 1 mM EDTA, 1% Triton-X100, and complete protease inhibitor cocktail (Roche)). Cell lysates were centrifuged and supernatants were precleared and immunoprecipitated with anti-FLAG M2 Affinity Gel (Sigma) at 4˚C with rocking for 4 h. Precipitates were then washed with lysis buffer and subjected to

sodium dodecyl sulfate-polyacrylamide gel electrophoresis (SDS-PAGE) separation. After SDS-PAGE, proteins were blotted onto PVDF membranes (Millipore). The blots were incubated in blocking buffer (5% fat-free milk in PBS buffer supplemented with 0.05% TWEEN-20) at room temperature (RT) for 1 h and incubated with primary antibodies in blocking buffer at 4°C for overnight. Blots were then washed and incubated in the horseradish peroxidase (HRP)-labeled secondary antibodies at RT for 1 h. After wash, blots were developed with ECL Prime Western Blotting System (GE Healthcare, RPN2232).

## Chromatin immunoprecipitation (ChIP) assay

The ChIP assay was performed as described previously [18,64]. Yeast cells in exponential growth phase were diluted to the same cell density, crosslinked for 20 min with 1% formaldehyde, quenched with 125 mM glycine for 10 min. Cells were pelleted and washed twice with 20 ml ice-cold PBS buffer and once with pre-chilled lysis buffer (50 mM HEPES, pH 7.5, 150 mM NaCl, 1 mM EDTA, 1% Triton-X100 and 0.1% sodium deoxycholate). The cell pellets were re-suspended in 500 μl lysis buffer containing 5 μl cocktail and 5 μl PMSF. Cells were lysed by using acid-washed glass beads, and then 250 μl of cell extracts were sonicated (pulse on 30 s, pulse off 30 s, 20 cycles) in a pre-chilled Bioruptor (Diagenode) to obtain chromatin fragments of about 300–500 bp in size. The soluble chromatin was obtained by centrifugation at full speed for 10 min. A 10 μl of the ChIP extract was taken for immunoprecipitation (IP) input, and 1.25 μl (1:200) of indicated antibodies (anti-Myc or anti-FLAG) were added to the remaining chromatin extract. Protein A Sepharose 4 Fast Flow beads (GE Healthcare) were washed three times with lysis buffer, and added to the ChIP extracts. After incubation at 4°C for 4–6 h, beads were washed once with lysis buffer, buffer I (50 mM HEPES, pH 7.5, 500 mM NaCl, 1 mM EDTA, 1% Triton-X100 and 0.1% sodium deoxycholate), buffer II (10 mM Tris-HCl, pH 8.0, 0.25 M LiCl, 1mM EDTA, NP-40 and 0.5% sodium deoxycholate) and TE (10 mM Tris-HCl, pH 8.0 and 1mM EDTA) each for 5 min. Bead-bound DNAs were eluted in 150 ul TE/1% SDS at 70°C for 30 min. IPs and inputs were incubated at 65°C overnight for reverse-crosslinking, and DNAs were purified with QIAquick PCR Purification Kit (QIAGEN). The real-time qPCR analysis was performed in the LightCycler 480 (Roche), and the TB Green *Premix Ex Taq* II (Tli RNaseH Plus) (TAKARA) reagent was used. The qPCR conditions were 30 s at 95°C, 40 cycles of 5 s at 95°C for denaturation, 30 s at 55°C for annealing and 30 s at 72°C for extension. Telomere enrichment was calculated as fold change of telomere product normalized to $act1^+$ locus product with the following formula $2^{[(Ct\ Act\ IP–Ct\ Act\ Input)-\ (Ct\ Tel\ IP–Ct\ Tel\ Input)]}$. The primers (JK380/381) targeting telomeres were used as described (S4 Table) [16].

Dot blot was performed as described previously [65]. Briefly, ChIP and input samples were denatured in denaturing solution (0.5 M NaOH, 1.5 M NaCl). After incubating at 55°C for 30 min, neutralizing solution (0.5 M Tris-HCl, 1.5 M NaCl) was added, and then all samples were transferred to a Hybond-N$^+$ Nylon membrane (GE Healthcare) by using the Bio-Dot Microfiltration Apparatus (BIO-RAD), UV cross-linked and incubated with Church buffer for 30 min at 50°C. Biotinylated telomeric-specific probe was incubated with the DNA at 50°C overnight, and biotin-probe-bound DNA fragments corresponding to telomeric DNAs were detected using Chemiluminescent Nucleic Acid Detection Module (Thermo Scientific, USA).

## Co-IP of TER1 RNA and Ccq1

The co-IP of TER1 and Ccq1 was performed as described [16,66]. Briefly, cells were pelleted and washed twice with 20 ml ice-cold PBS buffer and once with pre-chilled lysis buffer (50 mM HEPES, pH 7.5, 150 mM NaCl, 1 mM EDTA, 1% Triton-X100 and 0.1% sodium deoxycholate). The cell pellets were re-suspended in 500 μl lysis buffer containing 5 μl cocktail, 5 μl

PMSF, and 40 U/ml RNAase inhibitor. Cells were lysed by using acid-washed glass beads, and the supernatant was obtained by centrifugation at full speed for 10 min. Co-IP of Flag-tagged Ccq1 and TER1 was done with anti-Flag M2 antibody (Sigma) and protein A Sepharose beads (GE healthcare). The RNA on the beads was purified using RNeasy mini kit (QIAGEN), which was used as template for the reverse transcription using Prime Script RT reagent Kit with gDNA Eraser (Perfect Real Time) (TAKARA). The amount of TER1 was quantified using real-time qPCR analysis in the Light Cycler 480 (Roche), and the TB Green *Premix Ex Taq* II (Tli RNaseH Plus) (TAKARA) reagent was used. The qPCR conditions were 30 s at 95˚C, 40 cycles of 5 s at 95˚C for denaturation, 30 s at 55˚C for annealing and 30 s at 72˚C for extension. Primers for TER1 were used as described (S4 Table) [66]. % Precipitated TER1 RNA values were calculated based on ΔCt between Input and IP samples.

## Supporting information

**S1 Fig. Biochemical analysis of Pot1$_{DBD}$ binding to different telomeric ssDNAs.** (A) Gel filtration profile of Pot1$_{DBD}$ on a Superdex 200 column. The peak of Pot1$_{DBD}$ was resolved by SDS-PAGE and stained with Coomassie brilliant blue. (B-F) Gel filtration profiles of Pot1$_{DBD}$ binding to different telomeric ssDNAs. The A$_{260}$/A$_{280}$ ratios were calculated at the peak elution volume (V$_e$). (G) ITC measurements of interactions between Pot1$_{DBD}$ and telomeric ssDNAs. (H) Sequences of telomeric repeats used in the biochemical analysis.
(TIF)

**S2 Fig. Structural analyses of the Pot1$_{DBD}$-Tel18 interaction.** (A) Electron density map of Tel18 in the Pot1$_{DBD}$-Tel18 complex. Stereo view of the Sigma-A weighted 2F$_o$-F$_c$ map shows that Tel18 is well ordered in the crystal structure. Refined model of Tel18 is superimposed on the electron density map. Contours are drawn at the 1.0 σ level. (B) Electrostatic potential surface representation of the Pot1$_{DBD}$ protein. Positive potential, blue; negative potential, red (at the 10 kT $e^{-1}$ level). Tel18 is shown in stick model. (C) Schematic representation of the Pot1$_{DBD}$-Tel18 interaction. Cyan lines indicate hydrogen bonds and electrostatic interactions between Pot1$_{DBD}$ sidechains and ssDNA phosphates (circles) and bases (rectangles). Green lines indicate van der Waals contacts of Pot1$_{DBD}$ residues with bases as well as the stacking interactions between adjacent bases of Tel18. (D) Well-aligned C15-G16-Trp223-G18-Tyr224 stacking interaction in the Pot1$_{DBD}$-Tel18 structure. Sidechains of residues important for the interactions are shown in stick models. Dashed magenta lines denote the hydrogen-bonding interactions.
(TIF)

**S3 Fig. Multiple sequence alignment of Pot1 proteins from various fission yeast species.** Secondary structure elements of Pot1 are labeled on the top of the sequences. Three OB domains are boxed with respective colors as in Fig 1A. Conserved residues are boxed and highlighted in red. Red stars denote residues important for the stacking interactions observed in the Pot1$_{DBD}$-Tel18 crystal structure.
(TIF)

**S4 Fig. Pot1$_{DBD}$ can accommodate two telomeric repeats with a variable linker.** Structural modeling of Pot1$_{DBD}$ bound to two telomeric core repeats with zero (Tel15), one (Tel16) or two (Tel17) linker nucleotides based on the Pot1$_{DBD}$-Tel18 crystal structure.
(TIF)

**S5 Fig. Biochemical and structural analyses of the Pot1$_{OB3}$-Tpz1$_{PBM}$ interaction.** (A) Identification of the domains of Tpz1 and Pot1 that mediate the Pot1-Tpz1 interaction by Y2H

analysis. (B) Gel filtration chromatography profile of the $Pot1_{OB3}$-$Tpz1_{PBM}$ complex. The $Pot1_{OB3}$-$Tpz1_{PBM}$ complex fractions corresponding to the peak in the gel-filtration profile were resolved by SDS-PAGE and stained with Coomassie brilliant blue. (C) Multiple sequence alignment of Tpz1 proteins from various fission yeast species. Secondary structure elements of Tpz1 are labeled on the top of the sequences. The Pot1-, Ccq1- and Poz1-interacting motifs are indicted. Conserved residues are boxed and highlighted in red. (D) Electron density map of $Tpz1_{PIM}$ in the $Pot1_{OB3}$-$Tpz1_{PIM}$ complex. Stereo view of the Sigma-A weighted $2F_o$-$F_c$ map shows that $Tpz1_{PIM}$ is well ordered in the crystal structure. Refined model of $Tpz1_{PIM}$ is super-imposed on the electron density map. Contours are drawn at the $1.0\ \sigma$ level. (E) Tpz1 mutations that disrupt the Pot1-Tpz1 interaction have no effect on Tpz1-Ccq1 Y2H interactions. (TIF)

**S6 Fig. Structural comparison of the *S. pombe* Pot1-Tpz1, *O. nova* TEBPα-β and human POT1-TPP1 complexes.** (A) Superposition of *S. pombe* $Pot1_{OB3}$ and *O. nova* $TEBP\alpha_{OB3}$ crystal structures. (B) Superposition of *S. pombe* $Pot1_{OB3}$ and human $POT1_{OB3}$ crystal structures. (C-E) Structural comparison of the heterodimeric interactions in *S. pombe* Pot1-Tpz1 (C), *O. nova* TEBPα-β (D) and human POT1-TPP1 (E) complexes. (TIF)

**S7 Fig. Biochemical analyses of the $Tpz1_{CBM}$-$Ccq1_{TAD}$ interaction.** (A) Gel filtration chromatography profile of the $Tpz1_{CBM}$-$Ccq1_{TAD}$ complex. Elution positions of the 67 and 35 kDa protein markers are indicated. The $Tpz1_{CBM}$-$Ccq1_{TAD}$ complex fractions corresponding to the peak in the gel-filtration profile were resolved by SDS-PAGE and stained with Coomassie brilliant blue. (B) Multiple sequence alignment of Ccq1 proteins from various fission yeast species. Secondary structure elements of Ccq1 are labeled on the top of the sequences. Conserved residues are boxed and highlighted in red. (TIF)

**S8 Fig. Structural and mutational analyses of the $Tpz1_{CBM}$-$Ccq1_{TAD}$ interaction.** (A) Electron density map of the $Tpz1_{CBM}$ in the $Tpz1_{CBM}$-$Ccq1_{TAD}$ complex. Stereo view of the Sigma-A weighted $2F_o$-$F_c$ map shows that $Tpz1_{CBM}$ is well ordered in the crystal. Refined model of $Tpz1_{CBM}$ is superimposed on the electron density map. Contours are drawn at the $1.0\ \sigma$ level. (B) Superposition of the $Ccq1_{TAD}$ and *Sc*$Hda3_{NTD}$ (PDB: 3HGT) crystal structures. $Ccq1_{TAD}$ and *Sc*$Hda3_{NTD}$ are colored in slate blue and palegreen, respectively. (C) Electrostatic surface potential of the $Tpz1_{CBM}$-binding module on $Ccq1_{TAD}$ (positive potential, blue; negative potential, red). Two $Tpz1_{CBM}$ molecules are in ribbon representation and colored in orange and tint, respectively. (D) Tpz1 mutations that disrupt the Tpz1-Ccq1 interaction have no effect on Pot1-Tpz1 Y2H interactions. (TIF)

**S9 Fig. Functional analysis of the Tpz1-Ccq1 interaction.** (A) Telomere Southern blot analysis of the negative control *ccq1-L177R* cells. Genomic DNAs were digested with *Eco*R I and subjected to Southern blot analysis with a telomere-specific probe. (B) Effects of Ccq1 mutations on the transcriptional silencing of *his3*+ reporter gene inserted adjacent to the telomeric region. Equal amounts of 10-fold dilution series of cultures were spotted on YES or Pombe Medium Glutamate supplemented with uracil, leucine, and adenine (PMG ULA) (-histidine) plates. (C and D) Effects of the Ccq1-F155R mutation on telomere association for Ccq1, Tpz1, Rap1, Trt1, Clr3 and Clr4 were monitored by dot blot ChIP assays. (E) Co-IP of Ccq1 and *TER1 in vivo*. Data are represented as mean ± s.e.m. from four independent experiments. (TIF)

**S10 Fig. Architectural model of *S. pombe* shelterin complex.** Ribbon diagrams of the shelterin complex based on the atomic structures of $Pot1_{DBD}$-Tel18, $Pot1_{OB3}$-$Tpz1_{PIM}$, $Tpz1_{CBM}$-$Ccq1_{TAD}$, $Rap1_{PBM}$-Poz1-$Tpz1_{PBM}$ (PDB: 5XXF), $Rap1_{RCT}$-$Taz1_{RBM}$ (PDB: 2L3N) and $Taz1_{DD}$ (PDB: 4ZMK). Taz1 and Pot1 respectively bind to telomeric dsDNA and ssDNA regions via their Myb domains and OB folds, and are bridged by Rap1, Poz1 and Tpz1 via protein-protein interactions. The dimeric state of the shelterin complex is mediated by $Taz1_{DD}$, Poz1 and the $(Tpz1_{CBM}$-$Ccq1_{TAD})_2$ heterotetramer.
(TIF)

**S1 Table. Crystal data collection and refinement statistics.**
(XLSX)

**S2 Table. Statistical source data for Figs 4C, 4D, 4E, 4F, 4G, 4H, 4I and S9E.**
(XLSX)

**S3 Table. Yeast strains used for this study.**
(XLSX)

**S4 Table. Oligonucleotides used in this study.**
(XLSX)

## Author Contributions

**Conceptualization:** Ming Lei.

**Data curation:** Hong Sun, Zhenfang Wu, Yuanze Zhou, Zhixiong Zeng, Jian Wu.

**Funding acquisition:** Zhixiong Zeng, Jian Wu, Ming Lei.

**Investigation:** Hong Sun, Zhenfang Wu, Yuanze Zhou, Yanjia Lu, Huaisheng Lu, Hongwen Chen, Shaohua Shi.

**Methodology:** Hong Sun, Zhenfang Wu, Yuanze Zhou, Yanjia Lu, Huaisheng Lu.

**Supervision:** Ming Lei.

**Validation:** Zhenfang Wu, Jian Wu.

**Writing – original draft:** Zhenfang Wu, Jian Wu.

**Writing – review & editing:** Ming Lei.

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
