## [Decision Letter · Decision Letter 0]

20 Apr 2022

Dear Dr Lei,

Thank you very much for submitting your Research Article entitled 'Structural Insights into Fission Yeast Telomeric Overhang Binding Pot1-Tpz1-Ccq1 Complex' to PLOS Genetics.

The manuscript was fully evaluated at the editorial level and by independent peer reviewers. The reviewers appreciated the attention to an important problem, but raised some substantial concerns about the current manuscript. Based on the reviews, we will not be able to accept this version of the manuscript, but we would be willing to review a much-revised version. We cannot, of course, promise publication at that time.

If you decide to revise the manuscript for further consideration at PLOS Genetics, please aim to resubmit within the next 60 days, unless it will take extra time to address the concerns of the reviewers, in which case we would appreciate an expected resubmission date by email to plosgenetics@plos.org.

[LINK]

We are sorry that we cannot be more positive about your manuscript at this stage. Please do not hesitate to contact us if you have any concerns or questions.

Yours sincerely,

Hiroki Shibuya, PhD

Guest Editor

PLOS Genetics

John Greally

Section Editor: Epigenetics

PLOS Genetics

Reviewer's Responses to Questions

**Comments to the Authors:**

Reviewer #1: This study describes a series of new crystal structures of multiple protein-protein and protein-DNA interfaces within S. pombe shelterin. Specifically, the authors solve the structure of Pot1 DNA-binding domain with a long single stranded telomeric DNA sequence, Pot1-Tpz1 interface, and Tpz1-Ccq1 interface. The new Pot1-DNA structure recapitulates most of the past findings of the DNA-bound structures of individual OB domains while explaining how Pot1 can accommodate spacer sequences between the hexameric telomere repeats at chromosome ends. The Pot1-Tpz1 structure reveals similarities between how this interface is established in various model organisms including humans. The most novel structure described in the study is that of the “butterfly with antenna” shaped Tpz1-Ccq1 complex. The new protein-protein interfaces described in this study are validated using mutants analyzed by methods including yeast two hybrid, immunoprecipitation, telomere length analysis, and telomere ChIP. Overall there is a large amount of high-quality structural data in this manuscript that is backed up by biochemical validation. Although much of the results recapitulate previous findings and predictions based on other homologs, this is still an important contribution to model-system telomere structural biology. Below are major and minor points of critique.

Major points:

1. In multiple parts of the paper, the authors suggest that the Pombe Pot1 DNA binding mode is different from human because the human OB1-OB2 module is locked in one conformation. In a recent paper published in PLOS ONE, the Rhodes group has solved cryo-EM structures of human POT1-TPP1 to suggest that the OB1 and OB2 are not locked into a rigid body but can rather adopt different orientations about each other. Additionally, the same study showed how human POT1 can accommodate non-telomeric sequences between hexameric repeats, just like the pombe Pot1 protein. Unless the authors have a good reason to rebut these findings, the writing of the manuscript must take into consideration these recent data, which suggest that the human and Pombe OB1-OB2 modules are both flexible.

2. A justification is not provided for the kind of mutation that is made for the various interfaces. Why were Pot1 and Tpz1 residues always mutated to arginine in Fig 3G? Why were some of the Tpz1 mutations in Fig 5G alanine substitutions and other arginine substitutions? An arginine change would be expected to be more drastic, which would agree with the trend in the phenotype with these mutants.

3. In the yeast two hybrid (Y2H) assay in Fig. 3G, there is no evidence that the mutant protein is expressed or is folded? At least for the Tpz1 mutants, the authors could perform a Y2H against Ccq1 to show that the mutants are active in this biochemical function. This would provide strong evidence that the mutants are produced and are stably folded. In fact in Fig. 5H it seems like the two tested mutants are not expressed well. What is the expression level of the other mutants used in the Y2H in Fig. 5G? Y2H of these Tpz1 mutants with Pot1 should be performed to confirm that the mutants can still bind Pot1 as the mutations shouldn’t affect this interface.

4. Although the differences look large for the functional data in the manuscript, no statistical significance is provided. P-values should be reported for all effects that the authors propose are significant.

5. Figure 2 is largely based on modeling of shorter DNA substrates. However the method used for modeling was not described. This must be described in more detail, elaborating on whether energy minimization and/or geometry optimization was used in the protocol. Along the same line, the geometry (bond angles, bond lengths, dihedral angles) of the modeled nucleotides/linkages must be detailed along with how much they deviate from the standard values. While it is understandable that the authors did not solve new structures with each shorter DNA substrate, the modeling data should only be shown in the main figure if a rigorous computational method was used to generate the models. Otherwise, the modeling should be moved to the supplement. This is especially relevant to Tel15-bound structure, where the authors suggest a swinging away of the OB domains from each other.

6. Figure 3 is already dedicated to the Pot1-Tpz1 structure. It doesn’t seem justified to show superpositions of the same structure on other homologous structures as a new figure (figure 4). It seems most appropriate to move Figure 4 into the supplement or merge it with other panels in figure 3.

7. The authors make a point about differences in nt # 11, 12 and 13 between their structure and the OB2-9mer structure. What is the significance of this difference? Are the authors suggesting these are two alternative conformations or are they saying one is correct and the other isn’t? In either case, what is the physiological importance of attaining one conformation versus the other? Can the authors do additional experiments to determine which conformation is important for DNA binding affinity or dynamics or telomere function in vivo?

8. In general, the novelty of the Pot1-DNA structure is not described precisely. There are already two structures of Pot1 domains bound to DNA. Throughout the manuscript (including abstract, author summary, and multiple instances in the main text) it must be specified that the novelty of the structure presented here is the presence of both OB domains, and more importantly, the presence of the spacer sequences between hexameric repeats.

9. In multiple places in the manuscript (including abstract and main text), the authors seem to suggest that this study shows the structural basis of the heterochromatin function of Ccq1. However, this structural study does not have heterochromatin components like SHREC. The studies performed here show that the loss of the Tpz1-Ccq1 results in a loss of Ccq1 from telomeres, which results in heterochromatin defects. Thus the study is not showing how Ccq1 is performing its heterochromatin function. Instead it is showing that Ccq1 must be recruited to telomeres to perform its functions. This must be clarified and any overstatements of the facts revised.

Minor points:

1. Remove the word “While” at the beginning of the sentence in line 108; start directly with “Structures…”

2. In line 215, Asp415 is mentioned as part of a hydrophobic pocket. Either the term hydrophobic should be removed or toned down (mostly or largely hydrophobic) or Asp415 should not be listed there.

3. In line 31, it is most accurate to mention “DNA-bound” structures of POT1 domains.

4. In line 199: change to “Similar to how human POT1 interacts with TPP1”

5. In line 421 “constrain” should be “constraint”

6. In line 436 Figure 7 human shelterin diagram is not consistent with the current understanding of the stoichiometry, which has TRF2:RAP1 = 2:2.

Reviewer #2: In this ms by Sun et al, the authors determined 3 crystal structures, Pot1(DBD)-ss telomeric DNA, Pot1(OB3)-Tpz1(PIM), and Tpz1(CBM)-Ccq1(TAD). These are 3 individual interfaces in the telomere nucleoprotein complex, different from the comprehensive structure of the whole trimer as claimed by the authors in the title, “Telomeric Overhang Binding Pot1-Tpz1-Ccq1 Complex”. Pot1(DBD)-ss telomeric DNA is not entirely new because the structures of Pot1(OB1)-ssDNA and Pot1(OB2)-ssDNA structures were solved a while ago with Pot1(OB1)-ssDNA in 2003. In addition, the mechanism by which degenerate ss telomeric DNA seq is recognized was extensively elucidated by a serial of biochemical and biophysical work from the Baumann and Wuttke labs more than a decade ago. For the Pot1(OB3)-Tpz1(PIM) structure, its homologs in humans and in Ciliates have been solved and the work here presents little new structural insight. Tpz1(CBM)-Ccq1(TAD) is a new structure; however, the claimed new “functional insights into Ccq1-dependent telomere maintenance and telomeric heterochromatin formation” is merely repeat of previously published results using a slightly different mutants disrupting the same interface (as explained below).

Overall, this paper provides useful “for the record” structures of the fission yeast shelterin components, but significantly overclaims its new biological insight. It is a solid candidate for journals publishing protein structures (such as, Acta D), but lacks the level of biological insight and rigor in functional analysis for PLOS Genetics.

Major points:

1. Title “Structural Insights into Fission Yeast Telomeric Overhang Binding Pot1-Tpz1-Ccq1 Complex” misleads readers to think the paper solved the whole Pot1-Tpz1-Ccq1 complex structure. Need to change it to “Crystal structures of Pot1(DBD)-ss telomeric DNA, Pot1(OB3)-Tpz1(PIM), and Tpz1(CBM)-Ccq1(TAD)”.

2. “In this study, we determine the crystal structures of the Pot1-ssDNA, Pot1-Tpz1, and Tpz1-Ccq1 subcomplexes, providing not only structural basis for the telomeric overhang-binding module Pot1-Tpz1-Ccq1…” The authors must specify the residue range for each protein in the structures. Otherwise, it misleads the readers to think it is the full-length protein. Moreover, for “telomeric overhang-binding module Pot1-Tpz1-Ccq1”, it has no biological foundation to include Ccq1, because there is no evidence that Ccq1 works with Pot1-Tpz1 ssDNA binder. In fact, Ccq1 has more functional relationship to Poz1 because double deletion of poz1 and ccq1 causes telomere deprotection.

3. Pot1-Tpz1 is the complete ss telomeric DNA binder, not Pot1 by itself. Cech lab showed that Pot1-Tpz1 binds to ssDNA 10 times stronger than Pot1 by itself, indicating the contribution from Tpz1 to ssDNA binding just as ciliate TEBP-b does. The structural and biochemical study would generate new and complete picture of shelterin-ssDNA binding only if Pot1-Tpz1 complex is employed as the whole entity. To satisfy the authors’ claim of “telomeric overhang-binding module”, full length Pot1-Tpz1 complex needs to be characterized here structurally.

4. “Our biochemical analysis using purified proteins showed that a short and highly conserved fragment of Tpz1 (residues 185-212) is sufficient to maintain a stable interaction with Pot1-372-555”. The biochemical assay here (co-migration in gel filtration) only shows binding, but other parts of Tpz1 or Pot1 involved in the interaction might be omitted. To ensure that Tpz1-185-212 and Pot1-372-555 are the minimum but comprehensive interaction units for Pot1-Tpz1 interaction. The authors need to show that Tpz1-185-212 and Pot1-372-555 interaction has the same affinity as the full-length Pot1 and Tpz1.

5. The fission yeast molecular biology data (Fig 6) are merely repeats of similar data (disrupting the same interface using either the same or different mutants) and the data quality is poor. Barely any new biological insight was revealed from this figure, in contrast to what the authors claimed.

a. There is no molecular weight ladder, no labeling indicating which band represents telomeres. What is the identity of the top band? loading control? if so, which gene? Some other gene needs to be used for this purpose because it is too close to the telomere band. What is the difference between the two lanes of the same genetic background?

b. Based on the data in Fig. 5b, the degree of telomeric silencing does not correlate with the degree of Tpz1-Ccq1 disruption. For example, Ccq1-M147R mutant disrupts Tpz1-Ccq1 interaction as effective as L151R; however, M147R is almost at the WT level of silencing. In case like this, H3K9-CHIP, Western blot of major proteins involved (Tpz1, Ccq1, and Clr4) are required to sort things out.

c. The authors use Ccq1-TER1 RNA co-IP as a test to evaluate the role of Ccq1-Tpz1 interaction in telomerase recruitment. Due to complex recruitment pathway of telomerase to telomeres, the field standard is to use Trt1-CHIP to evaluate Trt1 recruitment in different genetic background.

d. The authors mentioned, “Notably, the amounts of telomeres associated Tpz1 and Rap1 were also decreased in the ccq1L151R and ccq1F155R cells (Fig. 5e, f), likely due to the telomere loss after the disruption of the Tpz1-Ccq1 interaction”. Telomere loss is normalized in the CHIP assay. The fact that the authors observed less telomere association of Tpz1 and Rap1 indicate that the telomere sequence is changed in the ccq1 mutant background. As Cooper lab (Tomita et al G&D 2008) showed that the cell maintains telomeres using the rad51-dependent recombination mechanism. Therefore, the seq composition of the telomeres is different from the telomerase-dependent mode. Slot-blot (instead of qPCR) must be used for the CHIP in this study (Fig 5d,e, f, h, i).

Minor points:

The authors need to give credit to other labs in the field by discussing the structures and results in the context of what was known. Specific areas include Pot1-Tpz1 and ssDNA binding work from the Wuttke, Cech, and Baumann labs, and Ccq1-Tpz1 interaction work from Nakamura and Jia labs.

Reviewer #3: 1) General comments

In this manuscript, Sun et al. have provided structural basis for the recognition of S. pombe degenerate telomeric sequences by Pot1 and the essential function of Ccq1 in telomere maintenance and telomeric heterochromatin formation by the determination of the crystal structures of the Pot1-ssDNA, Pot1-Tpz1 and Tpz1-Ccq1 subcomplexes. By this work, the authors are proposing an integrated model depicting how the S. pombe shelterin complex assembles and plays its roles at telomere. These findings are suggesting that the shelterin complex has evolved distinct molecular architectures to accommodate different functions in fission yeast and mammals during evolution. This study will help us to understand how the telomere is regulated by the shelterin complex in higher eukaryotes.

Most of the conclusions drawn by the authors are supported or suggested by the experimental data. The following points should be modified or answered:

2) Specific comments

Major points:

1. There is no interpretation for the results of Tpz1-W187R, Tpz1-N189R, and Tpz1-M190R mutants in Y2H in Fig. 3G.

2. ccq1-F177R mutant had better be included as a control in the analyses of Fig. 6A and B (or in supplemental Figures), because Ccq1-F177R mutation does not disturb Tpz1-Ccq1 interaction.

Minor points:

1. It seems to be difficult for readers to understand Fig.2A, so more detail descriptions or explanations are needed.

2. In page 16 lines 307 and 316, I am afraid that alpha 1 must be alpha 2, and alpha 2 must be alpha 3. Please confirm this point.

3. In page 26 lines 506-9, it is hard to understand why there is a description of GST-based purification here.

4. In page 29 lines 574, the reference #60 is not consistent with the description as “Moser et al, 2015” in S2 Table.

5. In page 29 lines 577, kanMx6 must be kanMX6.

6. In page 43 lines 949, EcoRI must be EcoRI(R should be non-italic).

7. the reference #60 is not consistent with the description as “Moser et al, 2015” in S2 Table.

**Have all data underlying the figures and results presented in the manuscript been provided?**

Reviewer #1: **No: **

Reviewer #2: None

Reviewer #3: Yes

PLOS authors have the option to publish the peer review history of their article (what does this mean?). If published, this will include your full peer review and any attached files.

Reviewer #1: No

Reviewer #2: No

Reviewer #3: **Yes: **Katsunori Tanaka

---

## [Decision Letter · Decision Letter 1]

22 Jun 2022

Dear Dr Lei,

We are pleased to inform you that your manuscript entitled "Structural Insights into Pot1-ssDNA, Pot1-Tpz1 and Tpz1-Ccq1 Interactions within Fission Yeast Shelterin Complex" has been editorially accepted for publication in PLOS Genetics. Congratulations!

One reviewer asked that you look carefully at Fig. S9B, as it appears that the two types of plates may be labeled incorrectly. In any finalisation of the manuscript you should address any such issues.

Yours sincerely,

Hiroki Shibuya, PhD

Guest Editor

PLOS Genetics

John Greally

Section Editor: Epigenetics

PLOS Genetics

Comments from the reviewers (if applicable):

Reviewer's Responses to Questions

**Comments to the Authors:**

Reviewer #1: The authors have carefully addressed the concerns raised in the first round of review. I do not have any further comments or suggestions for improvement. This manuscript is ready for publication in this reviewer’s opinion.

Reviewer #3: Everything except for the following point is answered well. The two type of plates of Fig.S9B seem to be labeled oppositely.

**Have all data underlying the figures and results presented in the manuscript been provided?**

Reviewer #1: Yes

Reviewer #3: Yes

PLOS authors have the option to publish the peer review history of their article (what does this mean?). If published, this will include your full peer review and any attached files.

Reviewer #1: No

Reviewer #3: **Yes: **Katsunori Tanaka

**Data Deposition**

http://datadryad.org/submit?journalID=pgenetics&manu=PGENETICS-D-22-00287R1

**Press Queries**

---

## [Editor Report · Acceptance letter]

12 Jul 2022

PGENETICS-D-22-00287R1 

Structural Insights into Pot1-ssDNA, Pot1-Tpz1 and Tpz1-Ccq1 Interactions within Fission Yeast Shelterin Complex 

Dear Dr Lei, 

We are pleased to inform you that your manuscript entitled "Structural Insights into Pot1-ssDNA, Pot1-Tpz1 and Tpz1-Ccq1 Interactions within Fission Yeast Shelterin Complex" has been formally accepted for publication in PLOS Genetics! Your manuscript is now with our production department and you will be notified of the publication date in due course.

With kind regards,

Zsofi Zombor

PLOS Genetics

On behalf of:
